

# Out-of-equilibrium dynamics of the XY spin chain from form factor expansion

**Etienne Granet⋆, Henrik Dreyer and Fabian H.L. Essler**

The Rudolf Peierls Centre for Theoretical Physics, Oxford University, Oxford OX1 3PU, UK

⋆ etienne.granet@physics.ox.ac.uk

## Abstract

We consider the XY spin chain with arbitrary time-dependent magnetic field and anisotropy. We argue that a certain subclass of Gaussian states, called Coherent Ensemble (CE) following [1], provides a natural and unified framework for out-of-equilibrium physics in this model. We show that *all* correlation functions in the CE can be computed using form factor expansion and expressed in terms of Fredholm determinants. In particular, we present exact out-of-equilibrium expressions in the thermodynamic limit for the previously unknown order parameter 1-point function, dynamical 2-point function and equal-time 3-point function.

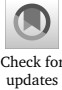
# 1 Introduction

Quantum integrable models are special models of many-body quantum physics with both a rich phenomenology and an exact Bethe-ansatz solution. But despite their "exact solvability", obtaining closed-form expressions in the thermodynamic limit for correlations of local observables in and out-of-equilibrium remains a formidable challenge. The standard approach to these problems [2] consists in expressing such correlation functions as form factor sums over the full Hilbert space. In interacting models, this task has been achieved only in certain parameter regimes, such as ground state correlations at late times and large distances [3–6], equal-time finite temperature correlations at short or large distances [7–16], full correlations in systematic strong coupling expansions [17, 18] or expansions in low densities of excitations [19–21], and also in some particularly simple interacting models or settings [22–25]. A number of numerical, approximate, field theory and other approaches aimed at facilitating form factor summations have been developed over the last decade and a half [26–42].

A subclass of quantum integrable models has arguably been of particular importance, namely theories that can be formulated in terms of free fermions. Examples include the Lieb-

Liniger model at infinite coupling [43,44] and the XY model in a field [45–50]. They constitute the point of departure and testbed of any field theory or exact method applying to the interacting case. But despite their free fermion formulation, the problem of obtaining analytic expressions for general in- and out-of-equilibrium correlations in the thermodynamic limit is still unsolved for some of these models. In fact, the computation of in- and out-of-equilibrium correlations can be said to be "fully" solved only for the models with a $U(1)$ symmetry such as the Lieb-Liniger model at infinite coupling and the XX chain. In this case, there exist integral, Pfaffian or Fredholm determinant representations for all static and dynamical correlations in arbitrary eigenstates [51–60], as well as for the full out-of-equilibrium time evolution of correlations after quantum quenches [56,61]. The exact tractability of the form factor expansion in these cases originates from the Cauchy determinant structure of the form factors [51].

However, there are still unknown correlation functions in the thermodynamic limit of free fermionic models without $U(1)$ symmetry such as the Transverse Field Ising Model and more generally the XY model in a field, despite a vast literature on the subject, see e.g. [45, 47, 49, 62–96]. While the quantities that are local in the underlying fermions (such as any correlation of the transverse magnetization, and the static $2n$-point functions of the order parameter) can be computed efficiently with Pfaffian representations arising from Wick's theorem [47, 49], there are no known exact representations for expectation values in general Gaussian states in the thermodynamic limit for the quantities that are non-local in the underlying fermions (such as static $(2n + 1)$-point functions of the order parameter, or any dynamical correlation of the order parameter). What makes the form factor expansion difficult to compute in these cases despite the model being free is that the form factors of the order parameter are not of Cauchy form. As a consequence, in terms of difficulty of the calculation these free models without $U(1)$ symmetry can be considered as in certain ways intermediate cases between free $U(1)$-symmetric and interacting models.

In contrast to the situation in the thermodynamic limit Pfaffian representations are readily available in finite systems with open boundary conditions, see e.g. [97]. Similarly, for finite systems with periodic boundary conditions one can invoke clustering properties [49, 50, 77, 79, 83] to obtain approximate representations. However, these representations typically scale with system size and as a far as we are aware their thermodynamic limits are generally not known.

In this work we show how to perform the form factor expansion for expectation values of *arbitrary* operators out of equilibrium. In particular, we derive the full time evolution of the order parameter one-point function, dynamical two-point function and static three-point function under arbitrary time-dependent ramps of the magnetic field and the anisotropy. We also derive alternative Fredholm determinant expressions for the full counting statistics of the transverse magnetization and order parameter two-point function using form factor expansions rather than Wick's theorem, hence with a method that is more generalizable to interacting models. This puts the XY model in a field on the same footing as the models with $U(1)$ symmetry with regards to out-of-equilibrium physics.

The technique we use to obtain these results is as follows. We define the *Coherent Ensemble* (CE) as the expectation value of operators within coherent states, which are superpositions of all zero-momentum pair states, weighted by amplitudes which are parameters of the CE. The crucial property of these coherent states is that they retain their structure when expressed in terms of eigenstates of the XY Hamiltonian with different values of magnetic field $h$ and anisotropy $\gamma$, as observed in [1] for the Ising model. This has two consequences: (i) The time evolution of the initial state with *any* variation of magnetic field $h(t)$ and anisotropy $\gamma(t)$ can be written as a coherent state with a certain amplitude; (ii) Any correlation function in the CE can be recast as a correlation function in an elementary (classical) Hamiltonian such as $-\sum_j \sigma_j^x \sigma_{j+1}^x$ for $h = 0, \gamma = 1$ or $-\sum_j \sigma_j^z$ for $h = \infty$. At these values of parameters, the

form factors of the order parameter are exactly Cauchy determinants, which enables one to use the techniques developed for $U(1)$ symmetric models and obtain Fredholm determinant expressions in the thermodynamic limit.

We note that these coherent states appeared more or less explicitly in different papers in the literature [67, 68, 74, 77, 98]. Most notably in [67, 68] they were used to obtain a Fredholm determinant expression for the two-point function of the order parameter at equilibrium at finite temperature. But to the best of our knowledge their utility in deriving Fredholm determinant representations for generic out-of-equilibrium correlators has not been realized prior to [1] and the present work.

The paper is organized as follows. We start by introducing coherent states in Section 10, and explain why out-of-equilibrium physics can be written as a CE. Then in Section 3 we show that arbitrary expectation values and correlation functions can be computed within the CE. Their derivation relies on a number of Lemmas for form factors and summation formulas that are gathered and proven in Appendix B. Finally, in Section 4 we apply our results to a number of examples including the Kibble-Zurek mechanism, Floquet physics and quantum quench physics.

## 2 Coherent Ensemble in the XY model

### 2.1 The XY model in a field

The Hamiltonian of the XY model on a system of size $L$ even, in a magnetic field $h$ and with anisotropy $\gamma$ is [45]

$$H(h,\gamma) = -\sum_{j=1}^{L} \frac{1+\gamma}{2}\sigma_j^x \sigma_{j+1}^x + \frac{1-\gamma}{2}\sigma_j^y \sigma_{j+1}^y + h\sigma_j^z. \tag{1}$$

We impose periodic boundary conditions $L+1 \equiv 1$. The diagonalisation of $H(h,\gamma)$ is reviewed in Appendix A. The Hamiltonian splits into two sectors $H(h,\gamma) = H^{\text{NS}}(h,\gamma) \oplus H^{\text{R}}(h,\gamma)$ called Neveu-Schwarz (NS) and Ramond (R) sector respectively

$$H^{\text{NS,R}}(h,\gamma) = \sum_{k \in \text{NS,R}} \varepsilon_{h\gamma}(k)\left(\alpha_{h\gamma;k}^\dagger \alpha_{h\gamma;k} - \frac{1}{2}\right), \tag{2}$$

where the fermions $\alpha_{h\gamma;k}$ satisfy canonical anti-commutation relations $\{\alpha_{h\gamma;k}, \alpha_{h\gamma;p}^\dagger\} = \delta_{k,p}$. Here, NS and R denote the sets

$$
\begin{aligned}
\text{NS} &= \left\{\frac{2\pi(n+1/2)}{L}, n = -L/2, \dots, L/2-1\right\}, \\
\text{R} &= \left\{\frac{2\pi n}{L}, n = -L/2, \dots, L/2-1\right\},
\end{aligned}
\tag{3}
$$

and $\varepsilon_{h\gamma}(k)$ denotes the energy of mode $k$

$$\varepsilon_{h\gamma}(k) = \begin{cases} 2\sqrt{(h-\cos k)^2 + \gamma^2 \sin^2 k} & \text{if } k \neq 0 \\ -2(1-h) & \text{if } k = 0 \end{cases}. \tag{4}$$

In these conventions, $\sigma^z$ (resp. $\sigma^x$) is local (resp. non-local) in the underlying Jordan-Wigner fermions[1], see Appendix A.

---

[1] We note that compared to the previous paper in Ising [1] the notations for $\sigma^x, \sigma^z$ have been switched, to match usual conventions in the quantum quench literature.

Denoting by $|0\rangle_{h\gamma}^{\text{NS,R}}$ the respective vacuum states annihilated by the $\alpha_{h\gamma;k}$'s in the NS and R sectors, the eigenstates of the model are then

$$
\begin{aligned}
|\boldsymbol{k}\rangle_{h,\gamma} &= \alpha_{h\gamma;k_1}^\dagger \dots \alpha_{h\gamma;k_N}^\dagger |0\rangle_{h\gamma}^{\text{NS}}, &\qquad \boldsymbol{k} &\subset \text{NS}, &\qquad N \text{ even}, \\
|\boldsymbol{k}\rangle_{h,\gamma} &= \alpha_{h\gamma;k_1}^\dagger \dots \alpha_{h\gamma;k_N}^\dagger |0\rangle_{h\gamma}^{\text{R}}, &\qquad \boldsymbol{k} &\subset \text{R}, &\qquad N \text{ odd}.
\end{aligned}
\tag{5}
$$

In these definitions we choose an ordering such that $k_i < k_j$ if $i < j$ and $k_i \neq 0, k_j \neq 0$. If $0 \in \boldsymbol{k}$ then we choose $k_N = 0$.

For $h > 1$, the ground state is $|0\rangle_{h\gamma}^{\text{NS}}$. For $0 < h < 1$ the two lowest energy states are $|0\rangle_{h\gamma}^{\text{NS}}$ and $\alpha_{h\gamma;0}^\dagger |0\rangle_{h\gamma}^{\text{R}}$. Their energy levels are exponentially close in $L$, and in finite size the true ground state is $|0\rangle_{h\gamma}^{\text{NS}}$. The model has two critical lines $|h| = 1, \gamma \neq 0$, and for $\gamma = 0, |h| < 1$ [47]. The energies of $|0\rangle_{h\gamma}^{\text{NS}}$ and $\alpha_{h\gamma;0}^\dagger |0\rangle_{h\gamma}^{\text{R}}$ are given by

$$
\begin{aligned}
\mathfrak{E}_{h\gamma}^{\text{NS}} &= -\sum_{k\in\text{NS}} \sqrt{(h-\cos k)^2 + \gamma^2 \sin^2 k}, \\
\mathfrak{E}_{h\gamma}^{\text{R}} &= -\sum_{k\in\text{R}} \sqrt{(h-\cos k)^2 + \gamma^2 \sin^2 k} + 2|1-h|\,\mathbf{1}_{h>1}.
\end{aligned}
\tag{6}
$$

Here we have defined

$$
\mathbf{1}_{h>1} = \begin{cases} 1 & \text{if } h > 1, \\ 0 & \text{else}. \end{cases}
\tag{7}
$$

## 2.2 Coherent states

We define $\text{NS}_+, \text{R}_+$ as the subsets of NS and R defined in (3) with strictly positive elements. Given $\boldsymbol{k} \subset \text{NS}_+$, we define pair states in the NS sector as the Fock states

$$
|\bar{\boldsymbol{k}}\rangle_{h\gamma}^{\text{NS}} = |\boldsymbol{k} \cup (-\boldsymbol{k})\rangle_{h\gamma}^{\text{NS}},
\tag{8}
$$

and given $\boldsymbol{k} \subset \text{R}_+$, pair states in the R sector as

$$
|\bar{\bar{\boldsymbol{k}}}\rangle_{h\gamma}^{\text{R}} = |\boldsymbol{k} \cup (-\boldsymbol{k}) \cup \{0\}\rangle_{h\gamma}^{\text{R}}.
\tag{9}
$$

Following [1], for a complex number $A$ called "phase" and a function $f$ called "amplitude", we introduce coherent states by

$$
\begin{aligned}
\Psi_{h\gamma}^{\text{NS}}(A,f) &\equiv A \sum_{\boldsymbol{k}\subset\text{NS}_+} \left[ \prod_{k\in\boldsymbol{k}} f(k) \right] |\bar{\boldsymbol{k}}\rangle_{h\gamma}^{\text{NS}} = A \prod_{k\in\text{NS}_+} \left[ 1 + f(k)\alpha_{h\gamma;-k}^\dagger \alpha_{h\gamma;k}^\dagger \right] |0\rangle_{h\gamma}^{\text{NS}}, \\
\Psi_{h\gamma}^{\text{R}}(A,f) &= A \sum_{\boldsymbol{k}\subset\text{R}_+} \left[ \prod_{k\in\boldsymbol{k}} f(k) \right] |\bar{\bar{\boldsymbol{k}}}\rangle_{h\gamma}^{\text{R}} = A \prod_{k\in\text{R}_+} \left[ 1 + f(k)\alpha_{h\gamma;-k}^\dagger \alpha_{h\gamma;k}^\dagger \right] \alpha_{h\gamma;0}^\dagger |0\rangle_{h\gamma}^{\text{R}}.
\end{aligned}
\tag{10}
$$

In these definitions the amplitude $f$ needs to be defined only on $[0, \pi]$. However we will consider it as an odd function defined on $[-\pi, \pi]$. We used the term "coherent" to denote these states in analogy with the better known coherent states for bosons $b$ defined by $e^{b^\dagger}|0\rangle$. The right-hand sides of (10) can indeed be written as the exponential of the bosonic terms $\alpha_{h\gamma;-k}^\dagger \alpha_{h\gamma;k}^\dagger$.

The key observation made in [1] for the transverse field Ising chain is the following relation between coherent states at different parameter values $(h, \gamma)$ and $(\tilde{h}, \tilde{\gamma})$:

**Theorem 1.** *Let $h, \tilde{h}$ and $\gamma, \tilde{\gamma}$ be arbitrary magnetic fields and anisotropies respectively. Then we have*

$$\Psi_{h\gamma}^{\mathrm{NS,R}}(A, f) = \Psi_{\tilde{h}\tilde{\gamma}}^{\mathrm{NS,R}}(\tilde{A}, \tilde{f}), \tag{11}$$

*where*

$$\tilde{A} = A \prod_{k \in \mathrm{NS}_+, \mathrm{R}_+} \frac{1 + i K_{\tilde{h}\tilde{\gamma}; h\gamma}(k) f(k)}{\sqrt{1 + K_{\tilde{h}\tilde{\gamma}; h\gamma}^2(k)}},$$

$$\tilde{f}(k) = \frac{i K_{\tilde{h}\tilde{\gamma}; h\gamma}(k) + f(k)}{1 + i K_{\tilde{h}\tilde{\gamma}; h\gamma}(k) f(k)}. \tag{12}$$

*Here we have defined*

$$K_{\tilde{h}\tilde{\gamma}; h\gamma}(k) = \tan \frac{\theta_k^{\tilde{h}\tilde{\gamma}} - \theta_k^{h\gamma}}{2}, \qquad e^{i\theta_k^{h\gamma}} = \frac{h - \cos k - i\gamma \sin k}{\sqrt{(h - \cos k)^2 + \gamma^2 \sin^2 k}}. \tag{13}$$

*Proof.* The proof is similar to that in [1] for the Ising model. Expanding the coherent state in a basis of energy eigenstates gives

$$\Psi_{h\gamma}^{\mathrm{NS}}(A, f) = A \sum_{\boldsymbol{q} \subset \mathrm{NS}} \sum_{\boldsymbol{r} \subset \mathrm{NS}_+} \left[ \prod_{r \in \boldsymbol{r}} f(r) \right] |\boldsymbol{q}\rangle_{\tilde{h}\tilde{\gamma}}^{\mathrm{NS}} \, {}_{\tilde{h}\tilde{\gamma}}^{\mathrm{NS}}\langle \boldsymbol{q} | \bar{\boldsymbol{r}} \rangle_{h\gamma}^{\mathrm{NS}}. \tag{14}$$

The overlaps ${}_{\tilde{h}\tilde{\gamma}}^{\mathrm{NS}}\langle \boldsymbol{q} | \bar{\boldsymbol{r}} \rangle_{h\gamma}^{\mathrm{NS}}$ between eigenstates of $H(h, \gamma)$ at different magnetic fields and anisotropies are given in Lemma 1 in Appendix B. Introducing the short-hand notation $K(k) \equiv K_{\tilde{h}\tilde{\gamma}; h\gamma}(k)$ we have

$$\Psi_{h\gamma}^{\mathrm{NS}}(A, f) = \frac{A}{\prod_{k \in \mathrm{NS}_+} \sqrt{1 + K^2(k)}} \sum_{\boldsymbol{q} \subset \mathrm{NS}_+} \left[ \prod_{q \in \boldsymbol{q}} [iK(q)] \sum_{\boldsymbol{r} \subset \mathrm{NS}_+} \prod_{r \in \boldsymbol{r}} \mathcal{F}(r, \boldsymbol{q}) \right] |\bar{\boldsymbol{q}}\rangle_{\tilde{h}\tilde{\gamma}}^{\mathrm{NS}},$$

$$\mathcal{F}(r, \boldsymbol{q}) = \begin{cases} \frac{f(r)}{iK(r)} & \text{if } r \in \boldsymbol{q}, \\ iK(r) f(r) & \text{if } r \notin \boldsymbol{q}. \end{cases} \tag{15}$$

The sum over $\boldsymbol{r}$ is

$$\sum_{\boldsymbol{r} \subset \mathrm{NS}_+} \prod_{r \in \boldsymbol{r}} \mathcal{F}(r, \boldsymbol{q}) = \prod_{q \in \boldsymbol{q}} \left( 1 + \frac{f(q)}{iK(q)} \right) \prod_{\substack{k \in \mathrm{NS}_+ \\ k \notin \boldsymbol{q}}} (1 + iK(k) f(k))$$

$$= \prod_{q \in \boldsymbol{q}} \frac{1 + \frac{f(q)}{iK(q)}}{1 + iK(q) f(q)} \prod_{k \in \mathrm{NS}_+} (1 + iK(k) f(k)). \tag{16}$$

Then

$$\Psi_{h\gamma}^{\mathrm{NS}}(A, f) = \tilde{A} \sum_{\boldsymbol{q} \subset \mathrm{NS}_+} \left[ \prod_{q \in \boldsymbol{q}} \tilde{f}(q) \right] |\bar{\boldsymbol{q}}\rangle_{\tilde{h}\tilde{\gamma}}^{\mathrm{NS}}, \tag{17}$$

with $\tilde{A}, \tilde{f}$ defined in the theorem. $\qquad \square$

## 2.3 Coherent Ensemble

The purpose of this section is to introduce the Coherent Ensemble which is convenient for formulating general time-dependent Hamiltonian dynamics.

### 2.3.1 Generalized Gibbs and Gaussian ensembles

We recall that the Generalized Gibbs Ensemble (GGE) parametrized by generalized temperatures $\beta_1, \beta_2, \ldots$ is defined by the following expectation values of an operator $\mathcal{O}$

$$\langle \mathcal{O} \rangle_{\boldsymbol{\beta}}^{\text{GGE}[h\gamma]} = \frac{\text{tr}\left[\mathcal{O}e^{-\sum_n \beta_n H_n}\right]}{\text{tr}\left[e^{-\sum_n \beta_n H_n}\right]}, \tag{18}$$

where $H_n$ are the conserved quantities of the model, and where tr denotes a trace over the full Hilbert space. These ensembles describe equilibrium physics (in the sense that local observables become time independent [99, 100]) in the XY model, be it finite-temperature equilibrium or steady states reached after a quantum quench. In the thermodynamic limit, they are equivalently parametrized by a particle density $\rho(\lambda)$ [101].

The Gaussian Ensemble (GE) parametrized by the $2 \times 2$ block $L \times L$ correlation matrix $\Gamma$ is defined by the fact that the expectation values satisfy Wick's theorem when expressed in terms of the Jordan-Wigner fermions $c_j$, see Appendix A, the elementary 2-point functions being given by

$$\Gamma_{ij} = \begin{pmatrix} \langle c_i^\dagger c_j \rangle_\Gamma^{\text{GE}[h\gamma]} & \langle c_i^\dagger c_j^\dagger \rangle_\Gamma^{\text{GE}[h\gamma]} \\ \langle c_i c_j \rangle_\Gamma^{\text{GE}[h\gamma]} & \langle c_i c_j^\dagger \rangle_\Gamma^{\text{GE}[h\gamma]} \end{pmatrix}. \tag{19}$$

GGE's are particular cases of GE's for the XY Hamiltonian (1).

### 2.3.2 Definition of the Coherent Ensemble

An operator $\mathcal{O}$ is called even (resp. odd) if its matrix elements between eigenstates with different (resp. same) fermion parity vanish. We define the Coherent Ensemble (CE) parametrized by an amplitude $f(k)$ by the following expectation values for even local operators $\mathcal{O}$

$$\langle \mathcal{O} \rangle_f^{\text{CE}[h\gamma]} = \Psi_{h\gamma}^{\text{NS}}(A^{\text{NS}}, f)^\dagger \mathcal{O} \Psi_{h\gamma}^{\text{NS}}(A^{\text{NS}}, f), \tag{20}$$

and for odd operators

$$\begin{aligned} \langle \mathcal{O} \rangle_f^{\text{CE}[h\gamma]} &= \frac{1}{2}\left[\Psi_{h\gamma}^{\text{R}}(A^{\text{R}}, f)^\dagger \mathcal{O} \Psi_{h\gamma}^{\text{NS}}(A^{\text{NS}}, f) + \Psi_{h\gamma}^{\text{NS}}(A^{\text{NS}}, f)^\dagger \mathcal{O} \Psi_{h\gamma}^{\text{R}}(A^{\text{R}}, f)\right] \\ &= \Re\left[\Psi_{h\gamma}^{\text{R}}(A^{\text{R}}, f)^\dagger \mathcal{O} \Psi_{h\gamma}^{\text{NS}}(A^{\text{NS}}, f)\right], \end{aligned} \tag{21}$$

where

$$A^{\text{NS,R}} = \prod_{k \in \text{NS}_+, \text{R}_+} (1 + |f(k)|^2)^{-\frac{1}{2}}. \tag{22}$$

Replacing NS by R in (20) incurs only negligible corrections in system size.

We note that the expression (21) naturally arises when the expectation value of an odd operator $\mathcal{O}$ is computed in a state that is a superposition of NS and R sector states

$$|\Psi\rangle = \frac{|\Psi^{\text{NS}}\rangle + |\Psi^{\text{R}}\rangle}{\sqrt{2}}. \tag{23}$$

### 2.3.3 CE as a particular case of a GE

Let us show that each CE corresponds to a particular GE. To that end, we consider the following expectation value of Jordan-Wigner fermions in momentum space

$$\langle c(k_1)^\dagger \ldots c(k_n)^\dagger c(q_1) \ldots c(q_m) \rangle_f^{\text{CE}[h\gamma]}, \tag{24}$$

with for example $k_1, \ldots, k_n, q_1, \ldots, q_m \in \mathrm{NS}$, and would like to show that it can be computed using Wick contractions. Using Theorem 1, we can write it in the $(\infty, \gamma)$ basis with another amplitude $f'$. In this basis we have

$$\langle c(k_1)^\dagger \ldots c(k_n)^\dagger c(q_1) \ldots c(q_m) \rangle_f^{\mathrm{CE}[h\gamma]} = \langle \alpha_{k_1}^\dagger \ldots \alpha_{k_n}^\dagger \alpha_{q_1} \ldots \alpha_{q_m} \rangle_{f'}^{\mathrm{CE}[\infty\gamma]}, \tag{25}$$

where the $\alpha$'s are implicitly written in the $(\infty, \gamma)$ basis for notational lightness. Next, we observe that for the expectation value $\langle \alpha_{k_1}^\dagger \ldots \alpha_{k_m}^\dagger \alpha_{q_1} \ldots \alpha_{q_m} \rangle_{f'}^{\mathrm{CE}[\infty\gamma]}$ to be non-zero, for each $k_i$ there has to be either another $k_j$ with $k_j = -k_i$, or a $q_j$ with $q_j = k_i$. The same holds true interchanging the $k$'s and the $q$'s. Hence we are led to evaluating expectation values of the form

$$\langle \prod_i \alpha_{k_i}^\dagger \alpha_{k_i} \prod_i \alpha_{-q_i}^\dagger \alpha_{q_i}^\dagger \prod_i \alpha_{r_i} \alpha_{-r_i} \prod_i \alpha_{-s_i}^\dagger \alpha_{s_i}^\dagger \alpha_{s_i} \alpha_{-s_i} \rangle_{f'}^{\mathrm{CE}[\infty\gamma]}, \tag{26}$$

where the $k$'s, $q$'s, $r$'s and $s$'s are all distinct. Using the definition of the CE, we obtain

$$\begin{aligned}
&\langle \prod_i \alpha_{k_i}^\dagger \alpha_{k_i} \prod_i \alpha_{-q_i}^\dagger \alpha_{q_i}^\dagger \prod_i \alpha_{r_i} \alpha_{-r_i} \prod_i \alpha_{-s_i}^\dagger \alpha_{s_i}^\dagger \alpha_{s_i} \alpha_{-s_i} \rangle_{f'}^{\mathrm{CE}[\infty\gamma]} \\
&= \prod_i \frac{|f'(k_i)|^2}{1 + |f'(k_i)|^2} \prod_i \frac{f'^*(q_i)}{1 + |f'(q_i)|^2} \prod_i \frac{f'(r_i)}{1 + |f'(r_i)|^2} \prod_i \frac{|f'(s_i)|^2}{1 + |f'(s_i)|^2}.
\end{aligned} \tag{27}$$

We now observe that the non-zero elementary two-point functions satisfy

$$\langle \alpha_k^\dagger \alpha_k \rangle_{f'}^{\mathrm{CE}[\infty\gamma]} = \frac{|f'(k)|^2}{1 + |f'(k)|^2}, \qquad \langle \alpha_k \alpha_{-k} \rangle_{f'}^{\mathrm{CE}[\infty\gamma]} = \frac{f'(k)}{1 + |f'(k)|^2}. \tag{28}$$

Because of the relation

$$\frac{|f'(s)|^2}{1 + |f'(s)|^2} = \langle \alpha_{-s}^\dagger \alpha_s^\dagger \rangle_{f'}^{\mathrm{CE}[\infty\gamma]} \langle \alpha_s \alpha_{-s} \rangle_{f'}^{\mathrm{CE}[\infty\gamma]} + \langle \alpha_{-s}^\dagger \alpha_{-s} \rangle_{f'}^{\mathrm{CE}[\infty\gamma]} \langle \alpha_s^\dagger \alpha_s \rangle_{f'}^{\mathrm{CE}[\infty\gamma]}, \tag{29}$$

we obtain that the right-hand side of (27) and so (24) can indeed be computed using Wick's theorem, which establishes that the CE is a particular case of a GE.

### 2.3.4 Inequivalence of CE with GE or GGE

CE ensembles are *not* equivalent to either GEs or GGEs. To show this, let us consider an operator $\mathcal{O}$ that is local in terms of the fermions $c_j$ and compute its expectation value within the CE. Using Wick's theorem in the thermodynamic limit it can be recast into sums and products of expectation values of quadratic terms in the Jordan Wigner fermions $c_j$'s in real space. These take the values

$$\begin{aligned}
\langle c_n^\dagger c_m \rangle_f^{\mathrm{CE}[h\gamma]} &= \frac{1}{2\pi} \int_{-\pi}^{\pi} e^{i(n-m)k} \left[ \cos^2(\theta_k^{h\gamma}/2) \frac{|f(k)|^2}{1 + |f(k)|^2} + \sin^2(\theta_k^{h\gamma}/2) \frac{1}{1 + |f(k)|^2} \right] dk \\
&\quad - \frac{1}{2\pi} \int_{-\pi}^{\pi} e^{i(n-m)k} \sin \theta_k^{h\gamma} \frac{\Im f(k)}{1 + |f(k)|^2} dk, \\
\langle c_n c_m \rangle_f^{\mathrm{CE}[h\gamma]} &= \frac{i}{4\pi} \int_{-\pi}^{\pi} e^{ik(n-m)} \sin \theta_k^{h\gamma} \frac{1 - |f(k)|^2}{1 + |f(k)|^2} dk \\
&\quad - \frac{1}{2\pi} \int_{-\pi}^{\pi} e^{ik(n-m)} \frac{\cos^2(\theta_k^{h\gamma}/2) f(k) + \sin^2(\theta_k^{h\gamma}/2) f^*(k)}{1 + |f(k)|^2} dk.
\end{aligned} \tag{30}$$

By computing the Fourier series, one sees that the values of $\langle c_n c_m \rangle_f^{\mathrm{CE}[h\gamma]}$ for all $n, m$ impose a system of two polynomial equations of degree 2 on $\Re f$ and $\Im f$. This prevents $\langle c_n^\dagger c_m \rangle_f^{\mathrm{CE}[h\gamma]}$

from taking arbitrary values, whereas in the GE they are independent quantities. Hence the CE's are a strict subset of GE's.

Expectation values of local operators in the thermodynamic limit of a GGE can be expressed in terms of mode occupation numbers or equivalently a root density $\rho$ [101] and the associated hole density $\rho_h = \frac{1}{2\pi} - \rho$ as

$$\langle c_n^\dagger c_m \rangle_\rho^{\text{GGE}[h\gamma]} = \int_{-\pi}^{\pi} e^{i(n-m)k} \cos^2(\theta_k^{h,\gamma}/2)\rho(k)\mathrm{d}k + \int_{-\pi}^{\pi} e^{i(n-m)k} \sin^2(\theta_k^{h\gamma}/2)\rho_h(-k)\mathrm{d}k \,,$$

$$\langle c_n c_m \rangle_\rho^{\text{GGE}[h\gamma]} = \frac{i}{2} \int_{-\pi}^{\pi} e^{ik(n-m)} \sin\theta_k^{h,\gamma} (\rho_h(-k) - \rho(k))\mathrm{d}k \,. \tag{31}$$

To have $\langle c_n c_m \rangle_\rho^{\text{GGE}[h\gamma]} = \langle c_n c_m \rangle_f^{\text{CE}[h\gamma]}$ for all $n,m$ requires a purely imaginary $f(k) \equiv i\tilde{f}(k)$ and the relation

$$\rho(k) = \frac{1}{2\pi} \frac{\tilde{f}(k)^2}{1+\tilde{f}(k)^2} + \frac{1}{2\pi \tan\theta_k^{h\gamma}} \frac{\tilde{f}(k)}{1+\tilde{f}(k)^2} \,. \tag{32}$$

The requirement that $\langle c_n^\dagger c_m \rangle_\rho^{\text{GGE}[h\gamma]} = \langle c_n^\dagger c_m \rangle_f^{\text{CE}[h\gamma]}$ for all $n,m$ further imposes that

$$\rho(k) = \frac{1}{2\pi} \frac{\tilde{f}(k)^2}{1+\tilde{f}(k)^2} - \frac{\tan\theta_k^{h\gamma}}{2\pi} \frac{\tilde{f}(k)}{1+\tilde{f}(k)^2} \,. \tag{33}$$

One sees that the two relations are compatible only if $\tilde{f}(k)$ takes the values 0 or $\infty$. In this case, the coherent state $\Psi_{h\gamma}^{\text{NS}}(A,f)$ is nothing but an eigenstate of the Hamiltonian. In fact, it is a "representative state" [101] of a root density that is either zero or maximal, which exactly corresponds to so-called "zero-entropy states", in the sense that their Yang-Yang entropy vanishes. Hence no GGE can be written as a CE, apart from zero-entropy state expectation values.

### 2.3.5 GGE at the boundary of CE

However, starting from a coherent state $\Psi_{h\gamma}^{\text{NS}}(f)$ one can obtain a GGE by taking the late time limit of the evolution of the CE induced by the Hamiltonian $H(h,\gamma)$. Indeed, one has

$$e^{-i\tau H(h,\gamma)} \Psi_{h\gamma}^{\text{NS,R}}(f,A) = \Psi_{h\gamma}^{\text{NS,R}}(f_\tau, A_\tau) \,, \tag{34}$$

with

$$f_\tau(k) = f(k)e^{-2i\tau\varepsilon_{h\gamma}(k)} \,, \qquad A_\tau = Ae^{-i\tau\mathfrak{E}_{h\gamma}^{\text{NS,R}}} \,. \tag{35}$$

Hence the CE after time $\tau$ is obtained from (30) by replacing $f$ by $f_\tau$. In the limit $\tau \to \infty$, the fast oscillations in $f_\tau(k)$ cause the second terms of the expectation values in (30) to vanish, while leaving the other terms invariant. This establishes that if the root density $\rho$ is even, and if one chooses $f$ such that

$$\rho(k) = \frac{1}{2\pi} \frac{|f(k)|^2}{1+|f(k)|^2} \,, \tag{36}$$

then

$$\lim_{\tau\to\infty} \langle c_n^\dagger c_m \rangle_{f_\tau}^{\text{CE}[h\gamma]} = \langle c_n^\dagger c_m \rangle_\rho^{\text{GGE}[h\gamma]} \,, \qquad \lim_{\tau\to\infty} \langle c_n c_m \rangle_{f_\tau}^{\text{CE}[h\gamma]} = \langle c_n c_m \rangle_\rho^{\text{GGE}[h\gamma]} \,. \tag{37}$$

This shows that any GGE with even root density can be obtained as a limit of CE. Figure 1 summarizes the inclusion of the different ensembles GE, CE and GGE.

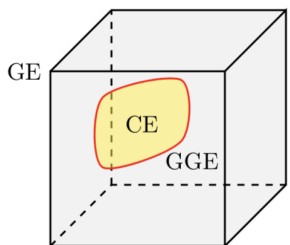

Figure 1: Sketch of the position of the GGE (red line) and CE (yellow surface) within the GE (gray volume), for symmetric root densities.

## 2.4 Out-of-equilibrium physics as a Coherent Ensemble

It is known that equilibrium physics can be formulated as a GGE [101]. The purpose of this section is to show how homogeneous non-equilibrium dynamics in the XY model can be formulated as a CE, see also Refs [71, 88, 98, 102, 103].

### 2.4.1 Differential equation for the amplitude

We assume that at time $t = 0$ the system is prepared in the ground state of $H(h_0, \gamma_0)$, and that it is time-evolved at time $t > 0$ with the Hamiltonian $H(h(t), \gamma(t))$, namely

$$|\psi(0)\rangle = |0\rangle_{h_0\gamma_0}^{NS},$$
$$i\partial_t |\psi(t)\rangle = H(h(t), \gamma(t))|\psi(t)\rangle. \tag{38}$$

We note that by virtue of the linearity of the Schrödinger equation one can equally well consider initial states that are superpositions, for example of the ground states in the NS and R sectors.

We would like to determine the time evolution of observables during this process. To that end, we replace the time evolution of the magnetic field and anisotropy by a series of quenches in which they are suddenly changed to $h_n = h(t_n), \gamma_n = \gamma(t_n)$ at times $t_n = (n-1)\delta t$, for a given small time interval $\delta t > 0$, and kept constant between these quenches. The original dynamics is obtained in the limit $\delta t \to 0$. We now observe that the initial state can be written as a coherent state

$$|\psi(0)\rangle = \Psi_{h_0\gamma_0}^{NS}(1, 0), \tag{39}$$

and that the time-evolution with $H(h, \gamma)$ of a coherent state written in the $(h, \gamma)$ basis is simply given by

$$e^{-itH(h,\gamma)}\Psi_{h\gamma}^{NS}(A, f) = \Psi_{h\gamma}^{NS}(A', f'), \tag{40}$$

with $f'(k) = f(k)e^{-2it\varepsilon_{h\gamma}(k)}$ and $A' = Ae^{-it\mathfrak{E}_{h\gamma}^{NS}}$. As a consequence, using repeatedly Theorem 1 to write the state as a coherent state in the $(h_n, \gamma_n)$ basis for $t_n \le t < t_{n+1}$, and then expressing it in the $(0, 1)$ basis, one has at time $t_n^-$

$$|\psi(t_n^-)\rangle = \Psi_{01}^{NS}(A_{(n-1)}^{NS}, f_{(n-1)}), \tag{41}$$

where the sequence of functions $f_j$ and phases $A_j$ satisfy

$$f_{(0)}(k) = -iK_{h_0\gamma_0;01}(k),$$

$$f_{(j)}(k) = \frac{iK_{h_j\gamma_j;01}(k)(e^{-2i\varepsilon_{h_j\gamma_j}(k)\delta t} - 1) + (K_{h_j\gamma_j;01}^2(k) + e^{-2i\varepsilon_{h_j\gamma_j}(k)\delta t})f_{(j-1)}(k)}{1 + e^{-2i\varepsilon_{h_j\gamma_j}(k)\delta t}K_{h_j\gamma_j;01}^2(k) + iK_{h_j\gamma_j;01}(k)(1 - e^{-2i\varepsilon_{h_j\gamma_j}(k)\delta t})f_{(j-1)}(k)},$$

$$A_j^{\mathrm{NS}} = A_{j-1}^{\mathrm{NS}} e^{-i\delta t \mathfrak{E}_{h_j\gamma_j}^{\mathrm{NS}}}$$

$$\times \prod_{k\in\mathrm{NS}_+} \frac{1 + K_{h_j\gamma_j;01}^2(k)e^{-2i\varepsilon_{h_j\gamma_j}(k)\delta t} + iK_{h_j\gamma_j;01}(k)f_{(j-1)}(k)(1 - e^{-2i\varepsilon_{h_j\gamma_j}(k)\delta t})}{1 + K_{h_j\gamma_j;01}^2(k)}. \tag{42}$$

We now take the limit $\delta t \to 0$. To that end it is useful to introduce a function $f_t(k)$ of both $t$ and $k$ by

$$f_t(k) = \lim_{\delta t\to 0} f_{(\lfloor t/\delta t\rfloor)}(k). \tag{43}$$

From (42), we conclude that the state of the system at time $t$ following an arbitrary variation $h(t), \gamma(t)$ of the magnetic field and anisotropy can be written as a coherent state

$$|\psi(t)\rangle = \Psi_{01}^{\mathrm{NS}}(A_t^{\mathrm{NS}}, f_t), \tag{44}$$

whose amplitude $f_t(k)$ satisfies a non-linear differential equation

$$\partial_t f_t(k) = \frac{2K_{h(t)\gamma(t);01}(k)}{1 + K_{h(t)\gamma(t);01}^2(k)}\varepsilon_{h(t)\gamma(t)}(k)(1 + f_t^2(k)) - 2i\frac{1 - K_{h(t)\gamma(t);01}^2(k)}{1 + K_{h(t)\gamma(t);01}^2(k)}\varepsilon_{h(t)\gamma(t)}(k)f_t(k). \tag{45}$$

The initial condition is $f_0(k) = -iK_{h(0)\gamma(0);01}(k)$. This shows that any expectation value out-of-equilibrium can be written as a CE. An example of the function $f_t(k)$ is plotted in Figure 2 for a sudden quench from $h_0 = 0.1$ to $h = 0.9$.

An equivalent system of linear differential equations was obtained previously in [98]. Indeed, we have

$$|\psi(t)\rangle = \prod_{k\in\mathrm{NS}_+} \left[ n_t(k) + m_t(k)\alpha_{01;-k}^\dagger \alpha_{01;k}^\dagger \right]|0\rangle_{01}^{\mathrm{NS}}, \tag{46}$$

where $n_t(p)$ an $m_t(p)$ fulfil the following system of linear ordinary differential equations

$$\frac{d}{dt}\begin{pmatrix} n_t(p) \\ m_t(p) \end{pmatrix} = \varepsilon_{h(t),\gamma(t)}(p)\begin{pmatrix} i\cos\Delta_t(p) & -\sin\Delta_t(p) \\ \sin\Delta_t(p) & -i\cos\Delta_t(p) \end{pmatrix}\begin{pmatrix} n_t(p) \\ m_t(p) \end{pmatrix}, \quad p \in \mathrm{NS}_+, \tag{47}$$

where we have defined

$$\Delta_t(k) = \theta_k^{h(t)\gamma(t)} - \theta_k^{01}, \tag{48}$$

and with the initial conditions

$$n_0(k) = \frac{1}{\sqrt{1 + K_{h_0\gamma_0;01}^2(k)}}, \quad m_0(k) = -\frac{iK_{h_0\gamma_0;01}(k)}{\sqrt{1 + K_{h_0\gamma_0;01}^2(k)}}. \tag{49}$$

This formulation is equivalent to (44) once we identify

$$f_t(p) = \frac{m_t(p)}{n_t(p)}, \quad A_t^{\mathrm{NS}} = \prod_{p\in\mathrm{NS}_+} n_t(p). \tag{50}$$

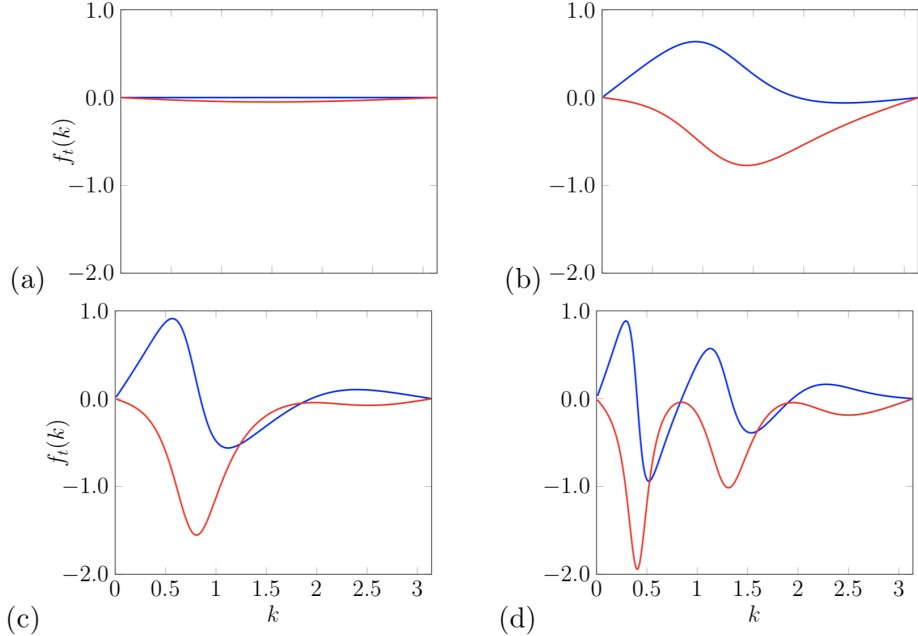

Figure 2: Amplitude $f_t(k)$ for a quantum quench from $h(0) = 0.1$ to $h(t) = 0.9$ for $t > 0$ at times (a) $t = 0$; (b) $t = 0.5$; (c) $t = 1$; (d) $t = 2$. The real and imaginary parts are shown in blue and red respectively.

### 2.4.2 The phase factor

In the limit $\delta t \to 0$, the phase becomes

$$
A_t^{\text{NS}} = A_0^{\text{NS}} e^{-i \int_0^t \mathfrak{E}_{h(s)\gamma(s)}^{\text{NS}} ds} \exp\left( \sum_{k \in \text{NS}_+} \varphi_t(k) \right),
$$

$$
\varphi_t(k) = -2 \int_0^t \varepsilon_{h(s)\gamma(s)}(k) \frac{K_{h(s)\gamma(s);01}(k)}{1 + K_{h(s)\gamma(s);01}^2(k)} (iK_{h(s)\gamma(s);01}(k) + f_s(k)) ds .
$$

(51)

For a coherent state in the R sector the same formula holds where the sum is over momenta in $\text{R}_+$ and with $\mathfrak{E}_{h(s)\gamma(s)}^{\text{NS}}$ replaced by $\mathfrak{E}_{h(s)\gamma(s)}^{\text{R}}$. For expectation values in the CE of even operators, the phase is irrelevant since it always cancels out. However, for odd operators the expectation value is proportional to the phase factor

$$
\phi_L(t) \equiv \frac{A_t^{\text{NS}}(A_t^{\text{R}})^*}{|A_t^{\text{NS}}(A_t^{\text{R}})^*|} ,
$$

(52)

where we made explicit the system size dependence of $\phi_L(t)$ that is only implicit in $A_t^{\text{NS,R}}$.

Let us assume first that the solution $f_t(k)$ to the non-linear differential equation (45) is

regular for all $t$ and $k$. Then, using the Euler-MacLaurin formula, we find

$$
\begin{aligned}
\sum_{k \in \mathrm{NS}_+} \varphi_t(k) &= \frac{L}{2\pi} \int_0^\pi \varphi_t(k) \mathrm{d}k + \mathcal{O}(L^{-1}), \\
\sum_{k \in \mathrm{R}_+} \varphi_t(k) &= \frac{L}{2\pi} \int_0^\pi \varphi_t(k) \mathrm{d}k - \frac{\varphi_t(\pi) + \varphi_t(0)}{2} + \mathcal{O}(L^{-1}), \\
\mathfrak{E}_{h\gamma}^{\mathrm{NS}} &= -\frac{L}{4\pi} \int_{-\pi}^\pi \varepsilon_{h\gamma}(k) \mathrm{d}k + \mathcal{O}(L^{-1}), \\
\mathfrak{E}_{h\gamma}^{\mathrm{R}} &= -\frac{L}{4\pi} \int_{-\pi}^\pi \varepsilon_{h\gamma}(k) \mathrm{d}k + 2|1-h|\mathbf{1}_{h>1} + \mathcal{O}(L^{-1}).
\end{aligned}
\tag{53}
$$

Assuming that the trajectory $h(t), \gamma(t)$ is such that the time spent on a critical point is of measure 0, we find $\varphi_t(\pi) = 0$ and

$$
\varphi_t(0) = -4i \int_0^t |1-h(s)| \mathbf{1}_{h(s)>1} \mathrm{d}s.
\tag{54}
$$

Hence in this case we obtain in the thermodynamic limit

$$
\phi_\infty(t) = 1.
\tag{55}
$$

However, if the function $f_t(k)$ is singular for some values $t^*, k^*$, then $\phi_L(t)$ for $t \geq t^*$ is not guaranteed to become 1 in the thermodynamic limit, which can result in a non-trivial multiplicative phase in (21). This phase has to be computed with (52) and (51).

Singularities of $f_t(k)$ are best understood with the system of linear differential equations (47). $n_t(p)$ and $m_t(p)$ are regular functions and the nature of the singularities of $f_t(p)$ becomes transparent: they simply correspond to situations when at least one probability amplitude $n_{t^*}(k^*)$ vanishes. This implies that the overlap of the time evolved state with $|0\rangle_{01}^{\mathrm{NS}}$ vanishes

$$
{}_{01}^{\mathrm{NS}}\langle 0| T \exp\left(-i \int_0^{t^*} H(h(s), \gamma(s)) \mathrm{d}s\right) |0\rangle_{h_0\gamma_0}^{\mathrm{NS}} = 0 \implies f_{t^*}(k^*) \text{ singular}.
\tag{56}
$$

This situation is somewhat reminiscent of non-analyticities in the Loschmidt amplitude [104]. This phase will be discussed again in a concrete example in Section 4.1.

To summarize this section, the CE provides the natural framework to evaluate the expectation value of any operator $\mathcal{O}$ during the out-of-equilibrium evolution (38). If $\mathcal{O}$ is even, then its expectation value is given by (20) with $f$ satisfying the nonlinear differential equation (45). If $\mathcal{O}$ is odd, then its expectation value is given by (21) multiplied (inside the real part) by the phase $\phi_L(t)$ (52). In the thermodynamic limit, $\phi_\infty(t)$ is constant in time as long as $f_t(k)$ is a regular function of $k$. If $f_{t^*}(k)$ has a singularity at $k^*$, then $\phi_\infty(t)$ can be discontinuous at $t^*$, and has to be evaluated according to (51).

## 3 Expectation values in the Coherent Ensemble

The purpose of this section is to show that essentially all correlation functions and expectation values in the CE can be expressed as Fredholm determinants and Pfaffians in the thermodynamic limit. We fix $h, \gamma$ and to ease notations write

$$
\langle \mathcal{O} \rangle_f \equiv \langle \mathcal{O} \rangle_f^{\mathrm{CE}[h\gamma]}.
\tag{57}
$$

### 3.1 Definitions

The formulas obtained for the various expectation values considered involve Fredholm determinants and Fredholm Pfaffians. In this section we present the definition of these objects and some of their properties.

#### 3.1.1 Fredholm determinant

Given a function $F(\lambda, \mu)$ on $[a, b] \times [a, b]$, the Fredholm determinant $\mathrm{Det}[\mathrm{Id} + F]$ is defined by

$$\mathrm{Det}[\mathrm{Id} + F] = 1 + \sum_{n=1}^{\infty} \frac{1}{n!} \int_a^b \cdots \int_a^b \det[F(z_i, z_j)]_{1 \le i, j \le n} \mathrm{d}z_1 \ldots \mathrm{d}z_n. \tag{58}$$

Here, Id should be merely considered as a notation. The Fredholm determinant satisfies the following relation

$$\mathrm{Det}[\mathrm{Id} + F] = \lim_{N \to \infty} \det\left[\delta_{i,j} + \frac{b-a}{N} F(\zeta_i, \zeta_j)\right]_{1 \le i, j \le N}, \tag{59}$$

with $\zeta_1 < \cdots < \zeta_N$ regularly spaced numbers covering $[a, b]$.

#### 3.1.2 Block Fredholm Pfaffian

Given a $2 \times 2$ matrix-valued function $\boldsymbol{K}(x, y) = (K_{ij}(x, y))_{1 \le i, j \le 2}$ on $[a, b] \times [a, b]$ satisfying $K_{ij}(x, y) = -K_{ji}(y, x)$, the block Fredholm Pfaffian $\mathrm{Pf}[\mathbf{Jd} + \boldsymbol{K}]$ is defined by [105]

$$\mathrm{Pf}[\mathbf{Jd} + \boldsymbol{K}] = 1 + \sum_{n=1}^{\infty} \frac{1}{n!} \int_a^b \cdots \int_a^b \mathrm{pf}[\boldsymbol{K}(z_i, z_j)]_{1 \le i, j \le n} \mathrm{d}z_1 \ldots \mathrm{d}z_n. \tag{60}$$

The matrices inside the Pfaffian on the right-hand side are thus $n \times n$ matrices of $2 \times 2$ blocks. Here, $\mathbf{Jd}$ should be merely considered as a notation. The block Fredholm Pfaffian satisfies the relation

$$\mathrm{Pf}[\mathbf{Jd} + \boldsymbol{K}] = \lim_{N \to \infty} \mathrm{pf}\left[\delta_{i,j}\boldsymbol{J} + \frac{b-a}{N}\boldsymbol{K}(\zeta_i, \zeta_j)\right]_{1 \le i, j \le N}, \tag{61}$$

with $J$ the $2 \times 2$ matrix

$$\boldsymbol{J} = \begin{pmatrix} 0 & 1 \\ -1 & 0 \end{pmatrix}. \tag{62}$$

#### 3.1.3 Fredholm Pfaffian

Given an antisymmetric function $F(\lambda, \mu)$ on $[-a, a] \times [-a, a]$, i.e. that satisfies $F(\mu, \lambda) = -F(\lambda, \mu)$, one can define the $2 \times 2$ matrix-valued function $\boldsymbol{K}_F$ on $[0, a] \times [0, a]$ by

$$\boldsymbol{K}_F(x, y) = \begin{pmatrix} F(x, y) & F(x, -y) \\ F(-x, y) & F(-x, -y) \end{pmatrix}. \tag{63}$$

We thus define the Fredholm Pfaffian $\mathrm{Pf}[\mathrm{Jd} + F]$ of an antisymmetric function $F$ on $[-a, a] \times [-a, a]$ by

$$\mathrm{Pf}[\mathrm{Jd} + F] \equiv \mathrm{Pf}[\mathbf{Jd} + \boldsymbol{K}_F]$$
$$= 1 + \sum_{n=1}^{\infty} \frac{1}{n!} \int_0^a \cdots \int_0^a \mathrm{pf}\begin{bmatrix} F(z_i, z_j) & F(z_i, -z_j) \\ F(-z_i, z_j) & F(-z_i, -z_j) \end{bmatrix}_{1 \le i, j \le n} \mathrm{d}z_1 \ldots \mathrm{d}z_n. \tag{64}$$

It satisfies the relation

$$\text{Pf}[\text{Jd} + F] = \lim_{\substack{N \to \infty \\ N \text{ even}}} (-1)^{N/2} \text{pf}\left[\delta_{i,N+1-j}\,\text{sgn}\,(j-i) + \frac{2a}{N}F(\zeta_i, \zeta_j)\right]_{1 \le i,j \le N}. \tag{65}$$

Here, $\zeta_1 < \cdots < \zeta_N$ are regularly spaced numbers covering $[-a, a]$ and assumed to be symmetrically distributed to ensure the antisymmetry of the matrix. The factor $(-1)^{N/2}$ compared to (61) arises from the re-ordering of rows and columns after changing the $2 \times 2$ block $N/2 \times N/2$ matrix into an $N \times N$ matrix, and re-ordering the negative $\zeta$'s in ascending order.

## 3.2 Full counting statistics of the transverse magnetization

As the operator $\sigma^z$ is local in the Jordan-Wigner fermions $c_j$, any static correlation of $\sigma^z$ is simple to calculate and can be expressed as a multiple integral in the thermodynamic limit. The purpose of this section is to derive a Fredholm determinant expression for the following generating function

$$\langle e^{i\theta \sum_{j=1}^{\ell} \sigma_j^z} \rangle_f, \tag{66}$$

for arbitrary $\theta$ and $\ell$. Exact Pfaffian representations of size $2\ell$ for the full counting statistics of the transverse magnetization in a generic GE have been derived before using Wick's theorem [71, 72, 85–87, 90, 106].

To compute (66), we express the coherent states involved in the CE in the $(\infty, \gamma)$ basis and expand them to obtain

$$\langle e^{i\theta \sum_{j=1}^{\ell} \sigma_j^z} \rangle_f = |A|^2 \sum_{\boldsymbol{\lambda},\boldsymbol{\mu} \subset \text{NS}_+} {}^{\text{NS}}_{\infty\gamma}\langle \bar{\boldsymbol{\lambda}} | e^{i\theta \sum_{j=1}^{\ell} \sigma_j^z} | \bar{\boldsymbol{\mu}} \rangle^{\text{NS}}_{\infty\gamma} \prod_{\lambda \in \boldsymbol{\lambda}} g^*(\lambda) \prod_{\mu \in \boldsymbol{\mu}} g(\mu), \tag{67}$$

where

$$g(k) = \frac{iK_{\infty\gamma;h\gamma}(k) + f(k)}{1 + iK_{\infty\gamma;h\gamma}(k)f(k)}. \tag{68}$$

We now use Lemma 2 to write the form factor of $e^{i\theta \sum_{j=1}^{\ell} \sigma_j^z}$ as a determinant

$$\langle e^{i\theta \sum_{j=1}^{\ell} \sigma_j^z} \rangle_f = e^{i\theta\ell}|A|^2 \sum_{\substack{\boldsymbol{\lambda},\boldsymbol{\mu} \subset \text{NS}_+ \\ |\boldsymbol{\lambda}|=|\boldsymbol{\mu}|}} \det E(\bar{\boldsymbol{\lambda}}, \bar{\boldsymbol{\mu}}) \prod_{\lambda \in \boldsymbol{\lambda}} g^*(\lambda) \prod_{\mu \in \boldsymbol{\mu}} g(\mu), \tag{69}$$

where $E(\boldsymbol{\lambda}, \boldsymbol{\mu})$ is defined in (179). Because of the pair structure of $\bar{\boldsymbol{\mu}}$ each $\mu_i$ appears in two columns of the matrix $E(\bar{\boldsymbol{\lambda}}, \bar{\boldsymbol{\mu}})$. Hence we can use Lemma 6 to carry out the sum over $\boldsymbol{\mu}$, with $\text{NS}_+$ being the set $K$, and (179) being the function $f$ when $\mu_k > 0$ and the function $g$ when $\mu_k < 0$. This gives

$$\sum_{\substack{\boldsymbol{\mu} \subset \text{NS}_+ \\ |\boldsymbol{\lambda}|=|\boldsymbol{\mu}|}} \det E(\bar{\boldsymbol{\lambda}}, \bar{\boldsymbol{\mu}}) \prod_{\mu \in \boldsymbol{\mu}} g(\mu) = (-1)^{N/2} \text{pf}[\tilde{E}(\bar{\boldsymbol{\lambda}}) - \tilde{E}(\bar{\boldsymbol{\lambda}})^T]. \tag{70}$$

In notations where $\bar{\boldsymbol{\lambda}} = \{\lambda_1, \ldots, \lambda_N\}$ the matrix elements of $\tilde{E}$ for $\lambda_j \neq -\lambda_k$ are given by

$$\tilde{E}(\bar{\boldsymbol{\lambda}})_{jk} = \left(\frac{e^{-2i\theta} - 1}{L}\right)^2 \sum_{\substack{\mu \in \text{NS}_+ \\ \mu \neq \lambda_j, -\lambda_k}} e^{i(\lambda_j + \lambda_k)} \frac{1 - e^{i\ell(\lambda_j - \mu)}}{1 - e^{i(\lambda_j - \mu)}} \frac{1 - e^{i\ell(\lambda_k + \mu)}}{1 - e^{i(\lambda_k + \mu)}} g(\mu)$$

$$+ (\mathbf{1}_{\lambda_j > 0} g(\lambda_j) + \mathbf{1}_{\lambda_k < 0} g(-\lambda_k))\left(1 + \frac{\ell}{L}(e^{-2i\theta} - 1)\right) \frac{e^{-2i\theta} - 1}{L} e^{i(\lambda_j + \lambda_k)} \frac{1 - e^{i\ell(\lambda_j + \lambda_k)}}{1 - e^{i(\lambda_j + \lambda_k)}}, \tag{71}$$

while for $\lambda_j = -\lambda_k$ we have

$$\tilde{E}_{jk}(\bar{\boldsymbol{\lambda}}) = \left(\frac{e^{-2i\theta}-1}{L}\right)^2 \sum_{\substack{\mu\in\text{NS}_+ \\ \mu\neq\lambda_j}} \left|\frac{1-e^{i\ell(\lambda_j-\mu)}}{1-e^{i(\lambda_j-\mu)}}\right|^2 g(\mu) + \mathbf{1}_{\lambda_j>0}\left(1+\frac{\ell}{L}(e^{-2i\theta}-1)\right)^2 g(\lambda_j).$$

(72)

The factor $(-1)^{N/2}$ arises from the re-ordering of the columns of the matrix in order to use Lemma 6. Factorizing $g(\lambda)$ for $\lambda > 0$, in the thermodynamic limit one obtains a Fredholm Pfaffian

$$(-1)^{N/2}\text{pf}[\tilde{E}(\bar{\boldsymbol{\lambda}}) - \tilde{E}(\bar{\boldsymbol{\lambda}})^T] = \text{Pf}[\text{Jd} + \mathcal{E}[\rho]]\prod_{\lambda\in\boldsymbol{\lambda}} g(\lambda)(1+o(L^0)).$$

(73)

Here the kernel acts on $[-\pi, \pi] \times [-\pi, \pi]$

$$\begin{aligned}
\mathcal{E}[\rho](\lambda, \mu) = {}& \frac{(e^{-2i\theta}-1)^2}{2\pi}\frac{\sqrt{\rho(\lambda)\rho(\mu)}}{g^+(\lambda)g^+(\mu)} \\
& \times \int_0^\pi \left[\frac{1-e^{i\ell(\lambda-k)}}{1-e^{i(\lambda-k)}}\frac{1-e^{i\ell(\mu+k)}}{1-e^{i(\mu+k)}} - \frac{1-e^{i\ell(\mu-k)}}{1-e^{i(\mu-k)}}\frac{1-e^{i\ell(\lambda+k)}}{1-e^{i(\lambda+k)}}\right] g(k)\text{d}k \\
& + (e^{-2i\theta}-1)\frac{\sqrt{\rho(\lambda)\rho(\mu)}}{g^+(\lambda)g^+(\mu)}\frac{1-e^{i\ell(\lambda+\mu)}}{1-e^{i(\lambda+\mu)}}(g(\lambda)-g(\mu)),
\end{aligned}$$

(74)

the function $\rho(\lambda)$ is the root density associated with $\boldsymbol{\lambda}$ and $g^+(\lambda)$ is defined by

$$g^+(\lambda) = \begin{cases} g(\lambda) & \text{if } \lambda > 0, \\ 1 & \text{if } \lambda < 0. \end{cases}$$

(75)

The factor $\sqrt{\rho(\lambda)\rho(\mu)}$ ensures that in the definition (64), each integral over $[0, \pi]$ comes with a root density factor $\rho(\lambda)$. Substituting (73) and (70) into (69) we obtain

$$\langle e^{i\theta\sum_{j=1}^\ell \sigma_j^z}\rangle_f = e^{i\theta\ell}|A|^2 \sum_{\boldsymbol{\lambda}\subset\text{NS}_+} \text{Pf}[\text{Jd} + \mathcal{E}[\rho]]\prod_{\lambda\in\boldsymbol{\lambda}} |g(\lambda)|^2(1+o(L^0)).$$

(76)

Finally we employ Lemma 7 to arrive at our final result in terms of a Fredholm Pfaffian

$$\boxed{\langle e^{i\theta\sum_{j=1}^\ell \sigma_j^z}\rangle_f = e^{i\theta\ell}\,\text{Pf}[\text{Jd} + \mathcal{E}[\rho_s]],}$$

(77)

where

$$\rho_s(k) = \frac{1}{2\pi}\frac{|g(k)|^2}{1+|g(k)|^2}.$$

(78)

## 3.3 Order-parameter one-point function

In contrast to $\sigma_\ell^z$ the longitudinal spin operator $\sigma_\ell^x$ is non-local in the Jordan-Wigner fermions and as a consequence the computation of its expectation value is a non-trivial problem. In this section we present a formula for the expectation value of the magnetization in the CE as defined in (21), i.e.

$$\langle\sigma_\ell^x\rangle_f \equiv \Re\left[\Psi_{h\gamma}^R(A^R, f)^\dagger \sigma_\ell^x \Psi_{h\gamma}^{NS}(A^{NS}, f)\right].$$

(79)

Since $\sigma_\ell^x$ is an odd operator under fermion parity it maps NS (R) states onto R (NS) states and only averages like (79) are non-vanishing. We note that they arise naturally in the context of spontaneous symmetry breaking of the spin-flip $\mathbb{Z}_2$ symmetry. The average (79) has been derived in the Supplemental Material of [1] in the particular case of the Ising model, and the

generalization to the XY model is straightforward. The result takes the form of a Fredhom determinant

$$\boxed{\langle \sigma_\ell^x \rangle_f = \Re \operatorname{Det}[\operatorname{Id} + \mathcal{M}[\rho_s]],} \tag{80}$$

where we defined the following kernel acting on $[0,\pi] \times [0,\pi]$

$$\mathcal{M}[\rho](\lambda,\mu) = -\frac{2}{\pi} \frac{\rho(\lambda)\sin\lambda}{h(\lambda)} \frac{1}{\cos\lambda - \cos\mu} \left[ \int_0^\pi \frac{h(k)\sin k}{\cos\lambda - \cos k} dk - \int_0^\pi \frac{h(k)\sin k}{\cos\mu - \cos k} dk \right], \tag{81}$$

$$h(k) = \frac{iK_{01;h\gamma}(k) + f(k)}{1 + iK_{01;h\gamma}(k)f(k)}, \qquad \rho_s(k) = \frac{1}{2\pi} \frac{|h(k)|^2}{1 + |h(k)|^2}. \tag{82}$$

In (80) we have assumed that the function $h$ in (82) is regular. We stress that in (79) the amplitudes $A^R$ and $A^{NS}$ are given by (22). In applications to time-dependent ramps the additional phase factor discussed in section 2.4.2 needs to be taken into account, *cf.* section 4.1.

## 3.4 Equal-time order-parameter two-point function

The purpose of this section is to derive the static two-point correlation function

$$\langle \sigma_{\ell+1}^x \sigma_1^x \rangle_f. \tag{83}$$

We note that exact Pfaffian/determinant representations of size $2\ell$ for the order parameter two point function in an arbitrary GE have been derived before using Wick's theorem and various explicit results on large-distance asymptotics have been derived, see e.g. Refs [45, 47, 49, 71, 76, 77].

To compute (83), we express the coherent states in the $(0,1)$ basis and insert a complete set of eigenstates between the two $\sigma^x$ operators to obtain

$$\langle \sigma_{\ell+1}^x \sigma_1^x \rangle_f = |A|^2 \sum_{\boldsymbol{\lambda},\boldsymbol{\mu} \subset NS_+} \sum_{\boldsymbol{\nu} \subset R} {}_{01}^{NS}\langle \bar{\boldsymbol{\lambda}} | \sigma_1^x | \boldsymbol{\nu} \rangle_{01}^R {}_{01}^R\langle \boldsymbol{\nu} | \sigma_1^x | \bar{\boldsymbol{\mu}} \rangle_{01}^{NS} \prod_{\lambda \in \boldsymbol{\lambda}} h^*(\lambda) \prod_{\mu \in \boldsymbol{\mu}} h(\mu) \prod_{\nu \in \boldsymbol{\nu}} e^{i\ell\nu}, \tag{84}$$

with $h(k)$ defined as in (82). Using Lemma 3 to express the form factor of $\sigma^x$ as a determinant, and Lemma 5 to sum over $\boldsymbol{\nu}$, we obtain

$$\langle \sigma_{\ell+1}^x \sigma_1^x \rangle_f = |A|^2 \sum_{\boldsymbol{\lambda},\boldsymbol{\mu} \subset NS_+} \det C(\bar{\boldsymbol{\lambda}}, \bar{\boldsymbol{\mu}}) \prod_{\lambda \in \boldsymbol{\lambda}} h^*(\lambda) \prod_{\mu \in \boldsymbol{\mu}} h(\mu), \tag{85}$$

where

$$C(\boldsymbol{p}, \boldsymbol{q})_{jk} = \frac{4}{L^2} \sum_{\nu \in R} \frac{e^{i(\ell+1)\nu}}{(e^{ip_j} - e^{i\nu})(e^{i\nu} - e^{iq_k})}. \tag{86}$$

To perform this sum, we now use Lemma 11. If $p_j \neq q_k$, we decompose the summand into partial fractions with respect to $e^{i\nu}$ and use (222) to carry out the sum over $\nu \in R$. If $p_j = q_k$ we use the derivative with respect to $z$ of (222). We obtain

$$C(\boldsymbol{p}, \boldsymbol{q})_{jk} = \begin{cases} -\frac{2}{L} \frac{e^{i\ell p_j} - e^{i\ell q_k}}{e^{ip_j} - e^{iq_k}} & \text{if } p_j \neq q_k, \\ \left(1 - \frac{2\ell}{L}\right) e^{ip_j(\ell-1)} & \text{if } p_j = q_k. \end{cases} \tag{87}$$

We next use Lemma 6 to sum over $\boldsymbol{\mu}$, which gives

$$\langle \sigma_{\ell+1}^x \sigma_1^x \rangle_f = |A|^2 \sum_{\boldsymbol{\lambda} \subset NS_+} (-1)^{N/2} \operatorname{pf}[\tilde{C}(\bar{\boldsymbol{\lambda}}) - \tilde{C}(\bar{\boldsymbol{\lambda}})^T] \prod_{\lambda \in \boldsymbol{\lambda}} h^*(\lambda). \tag{88}$$

Here $N$ is the number of momenta in $\bar{\boldsymbol{\lambda}}$ and

$$\tilde{C}(\boldsymbol{q})_{jk} = (1 - \delta_{q_j+q_k,0})\tilde{C}_1(q_j, q_k) + \delta_{q_j+q_k,0}\,\tilde{C}_2(q_j, q_k),$$

$$\tilde{C}_1(q_j, q_k) = \frac{4}{L^2} \sum_{\substack{p \in \mathrm{NS}_+ \\ p \neq q_j, -q_k}} \frac{e^{i\ell q_j} - e^{i\ell p}}{e^{iq_j} - e^{ip}} \frac{e^{-i\ell p} - e^{i\ell q_k}}{e^{-ip} - e^{iq_k}} h(p)$$

$$- (\mathbf{1}_{q_j>0} h(q_j) + \mathbf{1}_{q_k<0} h(-q_k)) \frac{2}{L}\left(1 - \frac{2\ell}{L}\right) \frac{1 - e^{i\ell(q_k+q_j)}}{1 - e^{i(q_k+q_j)}},$$

$$\tilde{C}_2(q_j, q_k) = \frac{4}{L^2} \sum_{\substack{p \in \mathrm{NS}_+ \\ p \neq q_j}} \left|\frac{e^{i\ell q_j} - e^{i\ell p}}{e^{iq_j} - e^{ip}}\right|^2 h(p) + \mathbf{1}_{q_j>0}\left(1 - \frac{2\ell}{L}\right)^2 h(q_j). \tag{89}$$

Taking the thermodynamic limit we obtain a Fredholm Pfaffian

$$(-1)^{N/2} \mathrm{pf}[\tilde{C}(\bar{\boldsymbol{\lambda}}) - \tilde{C}(\bar{\boldsymbol{\lambda}})^T] = \mathrm{Pf}[\mathrm{Jd} + \mathcal{C}_2[\rho]] \prod_{\lambda \in \boldsymbol{\lambda}} h(\lambda), \tag{90}$$

where $\rho$ is the root density corresponding to $\boldsymbol{\lambda}$ and where $\mathcal{C}_2[\rho]$ is the following kernel acting on $[-\pi, \pi] \times [-\pi, \pi]$

$$\mathcal{C}_2[\rho](\lambda, \mu) = -2\frac{\sqrt{\rho(\lambda)\rho(\mu)}}{h^+(\lambda)h^+(\mu)}\left[\frac{1 - e^{i(\lambda+\mu)\ell}}{1 - e^{i(\lambda+\mu)}}(h(\lambda) - h(\mu))\right.$$

$$\left. - \int_0^\pi \frac{\mathrm{d}k}{\pi}\left(\frac{1 - e^{i\ell(\lambda-k)}}{1 - e^{i(\lambda-k)}}\frac{1 - e^{i\ell(\mu+k)}}{1 - e^{i(\mu+k)}} - \frac{1 - e^{i\ell(\mu-k)}}{1 - e^{i(\mu-k)}}\frac{1 - e^{i\ell(\lambda+k)}}{1 - e^{i(\lambda+k)}}\right)h(k)\right],$$

with

$$h^+(\lambda) = \begin{cases} h(\lambda) & \text{if } \lambda > 0 \\ 1 & \text{if } \lambda < 0 \end{cases}. \tag{91}$$

We then employ Lemma 7 to arrive at our final result

$$\boxed{\langle \sigma^x_{\ell+1} \sigma^x_1 \rangle_f = \mathrm{Pf}[\mathrm{Jd} + \mathcal{C}_2[\rho_s]]}, \tag{92}$$

where

$$\rho_s(k) = \frac{1}{2\pi}\frac{|h(k)|^2}{1 + |h(k)|^2}. \tag{93}$$

### 3.5 Equal-time order-parameter three-point function

The purpose of this section is to show that the strategy employed for one and two-point functions can be generalized straightforwardly to higher-point functions. We consider the particular example of the order-parameter three-point function

$$\langle \sigma^x_{\ell_2+\ell_1+1} \sigma^x_{\ell_1+1} \sigma^x_1 \rangle_f. \tag{94}$$

This operator is odd and non-local in terms of the Jordan-Wigner fermions and as far as we are aware of there is no known Pfaffian or determinant representation of (94) in the thermodynamic limit.

We then follow the same steps as for the two-point function by expressing the two coherent states in the $(0, 1)$ basis and inserting complete sets of eigenstates between each operator to

obtain

$$\langle \sigma^x_{\ell_2+\ell_1+1} \sigma^x_{\ell_1+1} \sigma^x_1 \rangle_f = \Re A^{\mathrm{R}*} A^{\mathrm{NS}} \sum_{\substack{\boldsymbol{\lambda}\subset\mathrm{R}_+ \\ \boldsymbol{\mu}\subset\mathrm{NS}_+}} \sum_{\substack{\boldsymbol{\nu}\subset\mathrm{NS} \\ \boldsymbol{\kappa}\subset\mathrm{R}}} {}^{\mathrm{R}}_{01}\langle \bar{\boldsymbol{\lambda}}|\sigma^x_1|\boldsymbol{\nu}\rangle^{\mathrm{NSNS}}_{0101}\langle \boldsymbol{\nu}|\sigma^x_1|\boldsymbol{\kappa}\rangle^{\mathrm{R\;R}}_{0101}\langle \boldsymbol{\kappa}|\sigma^x_1|\bar{\boldsymbol{\mu}}\rangle^{\mathrm{NS}}_{01} $$
$$\times \prod_{\lambda\in\boldsymbol{\lambda}} h^*(\lambda) \prod_{\mu\in\boldsymbol{\mu}} h(\mu) \prod_{\nu\in\boldsymbol{\nu}} e^{i\ell_2\nu} \prod_{\kappa\in\boldsymbol{\kappa}} e^{i\ell_1\kappa}. \tag{95}$$

Next we perform the sum over $\boldsymbol{\kappa}$ by employing Lemmas 3 and 5 and obtain an analogous expression as in the two-point function case

$$\langle \sigma^x_{\ell_2+\ell_1+1} \sigma^x_{\ell_1+1} \sigma^x_1 \rangle_f = \Re A^{\mathrm{R}*} A^{\mathrm{NS}}$$
$$\times \sum_{\substack{\boldsymbol{\lambda}\subset\mathrm{R}_+ \\ \boldsymbol{\mu}\subset\mathrm{NS}_+}} \sum_{\boldsymbol{\nu}\subset\mathrm{NS}} {}^{\mathrm{R}}_{01}\langle \bar{\boldsymbol{\lambda}}|\sigma^x_1|\boldsymbol{\nu}\rangle^{\mathrm{NS}}_{01} \det C(\boldsymbol{\nu},\bar{\boldsymbol{\mu}}) \prod_{\lambda\in\boldsymbol{\lambda}} h^*(\lambda) \prod_{\mu\in\boldsymbol{\mu}} h(\mu) \prod_{\nu\in\boldsymbol{\nu}} e^{i(\ell_2-1/2)\nu}. \tag{96}$$

Here $C(\boldsymbol{\nu},\bar{\boldsymbol{\mu}})$ is given by (87) with $\ell$ replaced by $\ell_1$. Then we use Lemmas 3 and 5 to perform the sum over $\boldsymbol{\nu}$ and obtain

$$\langle \sigma^x_{\ell_2+\ell_1+1} \sigma^x_{\ell_1+1} \sigma^x_1 \rangle_f = \Re A^{\mathrm{R}*} A^{\mathrm{NS}} \sum_{\substack{\boldsymbol{\lambda}\subset\mathrm{R}_+ \\ \boldsymbol{\mu}\subset\mathrm{NS}_+}} \det C'(\bar{\boldsymbol{\lambda}},\bar{\boldsymbol{\mu}}) \prod_{\lambda\in\boldsymbol{\lambda}} h^*(\lambda) \prod_{\mu\in\boldsymbol{\mu}} h(\mu), \tag{97}$$

where

$$C'(\boldsymbol{p},\boldsymbol{q}) = \frac{2}{L} \sum_{\nu\in\mathrm{NS}} \frac{e^{i(\ell_2+1)\nu}}{e^{ip_j}-e^{i\nu}} \times \begin{cases} -\frac{2}{L}\frac{e^{i\ell_1\nu}-e^{i\ell_1 q_k}}{e^{i\nu}-e^{iq_k}} & \text{if } \nu\neq q_k, \\ (1-\frac{2\ell_1}{L})e^{i\nu(\ell_1-1)} & \text{if } \nu=q_k. \end{cases} \tag{98}$$

Writing

$$\frac{e^{i\ell_1\nu}-e^{i\ell_1 q_k}}{e^{i\nu}-e^{iq_k}} = e^{i(\ell_1-1)\nu} \sum_{m=0}^{\ell_1-1} e^{im(q_k-\nu)}, \tag{99}$$

we can use Eq (223) in Lemma 11 to compute $C'(\boldsymbol{p},\boldsymbol{q})$. We find

$$C'(\boldsymbol{p},\boldsymbol{q})_{jk} = \frac{2}{L} \frac{e^{i(\ell_1+\ell_2)p_j}-e^{i(\ell_1 q_k+\ell_2 p_j)}+e^{i(\ell_1+\ell_2)q_k}}{e^{ip_j}-e^{iq_k}}. \tag{100}$$

We then use Lemma 6 to sum over $\boldsymbol{\mu}$ to obtain

$$\langle \sigma^x_{\ell_2+\ell_1+1} \sigma^x_{\ell_1+1} \sigma^x_1 \rangle_f = \Re A^{\mathrm{R}*} A^{\mathrm{NS}} \sum_{\boldsymbol{\lambda}\subset\mathrm{R}_+} (-1)^{N/2} \mathrm{pf}[\tilde{C}'(\bar{\boldsymbol{\lambda}})-\tilde{C}'(\bar{\boldsymbol{\lambda}})^T] \prod_{\lambda\in\boldsymbol{\lambda}} h^*(\lambda), \tag{101}$$

where $N$ is the number of momenta in $\bar{\boldsymbol{\lambda}}$ and

$$\tilde{C}'(\bar{\boldsymbol{\lambda}})_{jk} = \frac{4}{L^2} \sum_{q\in\mathrm{NS}_+} \frac{e^{i(\ell_1+\ell_2)p_j}-e^{i(\ell_1 q+\ell_2 p_j)}+e^{i(\ell_1+\ell_2)q}}{e^{ip_j}-e^{iq}}$$
$$\times \frac{e^{i(\ell_1+\ell_2)p_k}-e^{i(-\ell_1 q+\ell_2 p_k)}+e^{-i(\ell_1+\ell_2)q}}{e^{ip_k}-e^{-iq}} h(q). \tag{102}$$

In the thermodynamic limit the remaining sum can be converted into an integral, except when $p_j = -p_q$ where an additional contribution $\delta_{p_j,-p_k} h(p_j)$ arises from the double pole in $q$. This results in a Fredholm Pfaffian

$$(-1)^{N/2} \mathrm{pf}[\tilde{C}'(\bar{\boldsymbol{\lambda}})-\tilde{C}'(\bar{\boldsymbol{\lambda}})^T] = \mathrm{Pf}[\mathrm{Jd}+\mathcal{C}_3[\rho]] \prod_{\lambda\in\boldsymbol{\lambda}} h(\lambda) + \mathcal{O}(L^{-1}), \tag{103}$$

where $\rho$ is the root density corresponding to $\boldsymbol{\lambda}$ and where $\mathcal{C}_3[\rho]$ is the following kernel acting on $[-\pi, \pi] \times [-\pi, \pi]$

$$\mathcal{C}_3[\rho](\lambda, \mu) = \frac{2}{\pi} \frac{\sqrt{\rho(\lambda)\rho(\mu)}}{h^+(\lambda)h^+(\mu)} \int_0^\pi \Big[ a(\lambda, k)\, a(\mu, -k) - a(\lambda, -k)\, a(\mu, k) \Big] h(k) \mathrm{d}k,$$

$$a(\lambda, k) = \frac{1 - e^{i\ell_2(\lambda-k)} + e^{i(\ell_1+\ell_2)(\lambda-k)}}{e^{i\lambda} - e^{ik}}. \tag{104}$$

This expression for $\mathcal{C}_3[\rho](\lambda, \mu)$ is to be understood as a principal value integral with simple poles at $k = \pm\lambda, \pm\mu$ for $\lambda \neq -\mu$, and is defined by continuity for $\lambda = -\mu$. Finally we apply Lemma 7 to (101), which results in the Fredholm Pfaffian

$$\boxed{\langle \sigma_{\ell_2+\ell_1+1}^x \sigma_{\ell_1+1}^x \sigma_1^x \rangle_f = \Re\, \mathrm{Pf}[\mathrm{Jd} + \mathcal{C}_3[\rho_s]],} \tag{105}$$

where

$$\rho_s(k) = \frac{1}{2\pi} \frac{|h(k)|^2}{1 + |h(k)|^2}. \tag{106}$$

In (105) we have once again assumed that the function $h$ in (82) is regular.

### 3.6 Dynamical order-parameter two-point function

We now turn to the non-equal-time two-point function of $\sigma^x$ in the CE, i.e.

$$C^{xx}(\ell, t) \equiv \langle \sigma_{\ell+1}^x(t/2)\sigma_1^x(-t/2)\rangle_f = \langle e^{itH(h,\gamma)/2}\sigma_{\ell+1}^x e^{-itH(h,\gamma)}\sigma_1^x e^{itH(h,\gamma)/2}\rangle_f. \tag{107}$$

A particular case of the correlator (107) is the dynamical two-point function in the XY model in a field after a quantum quench. This has been considered previously for $\gamma = 1$ and analytic results were obtained at low densities of excitations and large space/time separations [81].

#### 3.6.1 Summation of the $\sigma^x$ form factors

Without loss of generality we choose the coherent state in (107) to belong to the R sector and then expand it as (10) in the $(0,1)$ basis. We then insert a complete set of eigenstates between each of the operators to obtain

$$C^{xx}(\ell, t) = A_t^R A_{-t}^{R*} \sum_{\boldsymbol{q},\boldsymbol{k} \subset R_+} \sum_{\boldsymbol{\lambda},\boldsymbol{\mu} \subset NS} \left[ \prod_{q \in \boldsymbol{q}} h_{-t}^*(q) \prod_{k \in \boldsymbol{k}} h_t(k) \right]$$

$$\times \ _{01}^R \langle \bar{\bar{\boldsymbol{q}}} | \sigma_{\ell+1}^x | \boldsymbol{\lambda} \rangle_{01}^{NS} \ _{01}^{NS} \langle \boldsymbol{\lambda} | e^{-iH(h,\gamma)t} | \boldsymbol{\mu} \rangle_{01}^{NS} \ _{01}^{NS} \langle \boldsymbol{\mu} | \sigma_1^x | \bar{\bar{\boldsymbol{k}}} \rangle_{01}^R, \tag{108}$$

where we have from Theorem 1 and Eq (34)

$$h_t(k) = \frac{iK_{01;h\gamma}(k) + e^{it\varepsilon_{h\gamma}(k)}f(k)}{1 + iK_{01;h\gamma}(k)e^{it\varepsilon_{h\gamma}(k)}f(k)}, \tag{109}$$

$$A_t^R = e^{it\mathfrak{e}^R/2} \prod_{k \in R_+} \sqrt{\frac{1 + K_{01;h\gamma}(k)^2}{1 + |f(k)|^2}} \frac{1}{1 - iK_{01;h\gamma}(k)h_t(k)}. \tag{110}$$

For later convenience we introduce

$$A_t^{NS} = e^{it\mathfrak{e}^{NS}/2} \prod_{k \in NS_+} \sqrt{\frac{1 + K_{01;h\gamma}(k)^2}{1 + |f(k)|^2}} \frac{1}{1 - iK_{01;h\gamma}(k)h_t(k)}. \tag{111}$$

In the remainder of the section we will use the shorthand notations $K(k) \equiv K_{01;h\gamma}(k)$ and $\varepsilon(k) \equiv \varepsilon_{h\gamma}(k)$.

To evaluate (108), we first express the $\sigma^x$ form factors as determinants using Lemma 3. Because of the pair structure of the states $\bar{\bar{k}}$ and $\bar{\bar{q}}$, each $k_i$ and $q_j$ appear twice in these determinants. Hence the sums over $q, k \subset R_+$ are of the form of Lemma 6. It yields

$$
\begin{aligned}
C^{xx}(\ell, t) = A_t^R A_{-t}^{R*} \sum_{\boldsymbol{\lambda}, \boldsymbol{\mu} \subset \text{NS}} {}_{01}^{\text{NS}}\langle \boldsymbol{\lambda} | e^{-iH(h,\gamma)t} | \boldsymbol{\mu} \rangle_{01}^{\text{NS}} \; \text{pf}[D_t(\boldsymbol{\mu}) - D_t(\boldsymbol{\mu})^T] \\
\times \; \text{pf}[D_{-t}(\boldsymbol{\lambda}) - D_{-t}(\boldsymbol{\lambda})^T]^* \prod_{\lambda \in \boldsymbol{\lambda}} e^{i(\ell + 1/2)\lambda} \prod_{\mu \in \boldsymbol{\mu}} e^{-i\mu/2},
\end{aligned} \tag{112}
$$

where

$$
D_t(\boldsymbol{\mu})_{jk} = \frac{4}{L^2} \sum_{p \in R_+} \frac{h_t(p)}{(e^{ip} - e^{i\mu_j})(e^{-ip} - e^{i\mu_k})}. \tag{113}
$$

The thermodynamic limit of this expression is

$$
\begin{aligned}
D_t(\boldsymbol{\mu})_{jk} = & \; h_t(\mu_j) \delta_{\mu_j, -\mu_k} \mathbf{1}_{\mu_j > 0} \\
& + \frac{2}{\pi L(1 - e^{i(\mu_j + \mu_k)})} \left[ \int_0^\pi \frac{h_t(p)}{1 - e^{i(\mu_j - p)}} dp - \int_0^\pi \frac{h_t(p)}{1 - e^{-i(\mu_k + p)}} dp \right] + \mathcal{O}(L^{-2}),
\end{aligned} \tag{114}
$$

where the second term is understood as a derivative when $\mu_j = -\mu_k$.

### 3.6.2 Thermodynamic limit of the Pfaffians

The thermodynamic limit of the Pfaffians appearing in (112) is more involved than for the equal-time correlations treated in the previous sections. Indeed, $\boldsymbol{\lambda}$ and $\boldsymbol{\mu}$ are not necessarily pair states and so the "anti-diagonal" term $\delta_{\mu_j, -\mu_k} \mathbf{1}_{\mu_j > 0}$ in (114) is not always present. To treat this complication we introduce two sets of momenta $\pi(\boldsymbol{\mu}), \sigma(\boldsymbol{\mu})$ as in Lemma 1. One sees that the behaviour of $D_t(\boldsymbol{\mu}) - D_t(\boldsymbol{\mu})^T$ significantly depends on whether the $\mu$'s are paired $\mu \in \pi(\boldsymbol{\mu})$, in which case there is a non-zero anti-diagonal term $\delta_{\mu_j, -\mu_k}$ of order $L^0$, or whether they are not paired $\mu \in \sigma(\boldsymbol{\mu})$, in which case this "anti-diagonal" term is absent. In order to use Lemma 10 we employ Cayley's relation

$$
\text{pf}[D_t(\boldsymbol{\mu}) - D_t(\boldsymbol{\mu})^T]^2 = \det[D_t(\boldsymbol{\mu}) - D_t(\boldsymbol{\mu})^T], \tag{115}
$$

and write

$$
\begin{aligned}
[D_t(\boldsymbol{\mu}) - D_t(\boldsymbol{\mu})^T]_{jk} = & \; h_t(\mu_j) \delta_{\mu_j, -\mu_k} + \frac{1}{L} d_t(\mu_j, \mu_k), \\
d_t(\lambda, \mu) = & \; \frac{2}{\pi(1 - e^{i(\lambda + \mu)})} \left[ \int_0^\pi \frac{h_t(p)}{1 - e^{i(\lambda - p)}} dp + \int_0^\pi \frac{h_t(p)}{1 - e^{-i(\lambda + p)}} dp \right. \\
& \left. - \int_0^\pi \frac{h_t(p)}{1 - e^{i(\mu - p)}} dp - \int_0^\pi \frac{h_t(p)}{1 - e^{-i(\mu + p)}} dp \right].
\end{aligned} \tag{116}
$$

In the determinant (115) we then rearrange the lines and columns in such a way that the paired momenta $\mu_j \in \pi(\boldsymbol{\mu})$ appear on the "anti-diagonal" of the matrix $D_t(\boldsymbol{\mu}) - D_t(\boldsymbol{\mu})^T$ and are ordered among themselves (but the unpaired momenta in $\sigma(\boldsymbol{\mu})$ are not necessarily ordered). We then factorize $D_t(\boldsymbol{\mu}) - D_t(\boldsymbol{\mu})^T = LR$, where

$$
R_{ij} = \delta_{i, N+1-j} \, \text{sgn}(j - i) \prod_{\mu \in \pi(\boldsymbol{\mu})} h_t^2(\mu), \quad i, j = 1, \dots, N. \tag{117}
$$

This way, the determinant $\det[D_t(\boldsymbol{\mu})-D_t(\boldsymbol{\mu})^T]$ is of the form of Lemma 10, with $n$ the number of unpaired momenta $\sigma(\boldsymbol{\mu})=\{v_1,\ldots,v_n\}$ and with functions

$$f(\lambda,\mu)=\frac{d_t(\lambda,-\mu)}{h_t^+(\lambda)h_t^+(\mu)},\quad g_j(\lambda)=\frac{d_t(\lambda,v_j)}{h_t^+(\lambda)},\quad h_i(\mu)=\frac{d_t(v_i,-\mu)}{h_t^+(\mu)},\quad a_{i,j}=d_t(v_i,v_j),\tag{118}$$

where we introduced

$$h_t^+(\lambda)=\begin{cases}h_t(\lambda)&\text{if }\lambda>0,\\1&\text{if }\lambda<0.\end{cases}\tag{119}$$

We thus obtain as we approach the thermodynamic limit

$$\det[D_t(\boldsymbol{\mu})-D_t(\boldsymbol{\mu})^T]=\frac{1}{L^{|\sigma(\boldsymbol{\mu})|}}\mathrm{Det}_{\lambda,\mu}[\mathrm{Id}+\mathcal{D}_{\rho,t}(\lambda,-\mu)]$$
$$\times\det\left[\mathcal{F}_{\rho,t}(v_i,v_j)\right]_{v_i,v_j\in\sigma(\boldsymbol{\mu})}\prod_{\mu\in\pi(\boldsymbol{\mu})}h_t^2(\mu),\tag{120}$$

where $\rho$ the root density corresponding to $\boldsymbol{\mu}$, and where we defined the following kernel acting on $[-\pi,\pi]\times[-\pi,\pi]$

$$\mathcal{D}_{\rho,t}(\lambda,\mu)=\frac{\sqrt{\rho(\lambda)\rho(\mu)}}{h_t^+(\lambda)h_t^+(\mu)}d_t(\lambda,\mu),\tag{121}$$

and $\mathcal{F}_{\rho,t}(\lambda,\mu)$ satisfies the linear integral equation

$$\mathcal{F}_{\rho,t}(\lambda,\mu)+\int_{-\pi}^{\pi}\frac{d_t(\lambda,-v)}{h_t^+(\lambda)h_t^+(v)}\mathcal{F}_{\rho,t}(v,\mu)\rho(v)\mathrm{d}v=d_t(\lambda,\mu).\tag{122}$$

Eqn. (122) is obtained from (208) by using the equation for the resolvent (209) as well as its equivalent definition (219). It is useful to define two further functions by

$$\mathcal{D}'_{\rho,t}(\lambda,\mu)=\frac{\sqrt{\rho(\lambda)\rho(\mu)}}{(h_t^+)^*(\lambda)(h_t^+)^*(\mu)}d_t^*(\lambda,\mu),$$
$$\mathcal{F}'_{\rho,t}(\lambda,\mu)+\int_{-\pi}^{\pi}\frac{d_t^*(\lambda,-v)}{(h_t^+)^*(\lambda)(h_t^+)^*(v)}\mathcal{F}'_{\rho,t}(v,\mu)\rho(v)\mathrm{d}v=d_t^*(\lambda,\mu).\tag{123}$$

Now, using that $d_t(\lambda,\mu)=-d_t(\mu,\lambda)$, we find from (122) and (219) that $\mathcal{F}_{\rho,t}(\lambda,\mu)=-\mathcal{F}_{\rho,t}(\mu,\lambda)$. Hence $\left[\mathcal{F}_{\rho,t}(v_i,v_j)\right]_{v_i,v_j\in\sigma(\boldsymbol{\mu})}$ is an antisymmetric matrix, and we can write its determinant as the square of its Pfaffian

$$\det\left[\mathcal{F}_{\rho,t}(v_i,v_j)\right]_{v_i,v_j\in\sigma(\boldsymbol{\mu})}=\mathrm{pf}\left[\mathcal{F}_{\rho,t}(v_i,v_j)\right]_{v_i,v_j\in\sigma(\boldsymbol{\mu})}^2.\tag{124}$$

This results in a Fredholm Pfaffian as we approach the thermodynamic limit

$$(-1)^{N/2}\mathrm{pf}[D_t(\boldsymbol{\mu})-D_t(\boldsymbol{\mu})^T]=\frac{\mathfrak{s}(\sigma(\boldsymbol{\mu}),t)}{L^{|\sigma(\boldsymbol{\mu})|/2}}\mathrm{Pf}[\mathrm{Jd}+\mathcal{D}_{\rho,t}]\tag{125}$$
$$\times\mathrm{pf}\left[\mathcal{F}_{\rho,t}(v_i,v_j)\right]_{v_i,v_j\in\sigma(\boldsymbol{\mu})}\prod_{\mu\in\pi(\boldsymbol{\mu})}h_t(\mu).\tag{126}$$

Here, $\mathfrak{s}(\sigma(\boldsymbol{\mu}),t)$ is a function that takes the values $\pm1$. Let us argue that it is always equal to 1. By definition of the Fredholm Pfaffian, we know from (65) that $\mathfrak{s}(\{\},t)=1$ indeed. Besides, expanding the Pfaffian on the left-hand side of (125) on the lines and columns where the elements of $\sigma(\boldsymbol{\mu})$ are present, we see that it is an integral over a finite product of terms

$d_t(\mu, \nu)$ for $\nu \in \sigma(\boldsymbol{\mu})$, which is regular in $\nu$ assuming $h_t$ is regular. Hence $\mathfrak{s}(\sigma(\boldsymbol{\mu}), t)$ cannot depend on the elements of $\sigma(\boldsymbol{\mu})$. However it can still depend a priori on the number of unpaired momenta $|\sigma(\boldsymbol{\mu})|$ as well as on $t$. To go further, let us consider the antisymmetric matrix $B(\boldsymbol{\kappa})_{jk} = h_t(\kappa_j)\delta_{\kappa_j, -\kappa_k} + \frac{1+\epsilon}{L}d_t(\kappa_j, \kappa_k)$ for a paired state $\boldsymbol{\kappa}$ and a small $\epsilon$. By expanding the Pfaffian, we have

$$\mathrm{pf}B(\boldsymbol{\kappa}) = \sum_{\substack{J \subset \boldsymbol{\kappa} \\ |J| \text{ even}}} \epsilon^{|J|/2}\, \mathrm{pf}B_J(\boldsymbol{\kappa}), \tag{127}$$

with $B_J(\boldsymbol{\kappa})_{jk} = h_t(\kappa_j)\delta_{\kappa_j, -\kappa_k}\mathbf{1}_{\kappa_j \notin J} + \frac{1}{L}d_t(\kappa_j, \kappa_k)$. In this sum, there are $\mathcal{O}(L^{2n})$ elements for $|J| = 2n$, whereas the Pfaffian $\mathrm{pf}B_J(\boldsymbol{\kappa})$ is of order $\mathcal{O}(L^{-m})\mathrm{pf}B(\boldsymbol{\kappa})$ where $m$ is the number of momenta either that are in $J$ or whose opposite is in $J$ (or both). Indeed, in these cases the $\mathcal{O}(L^0)$ anti-diagonal term $\delta_{\kappa_j, -\kappa_k}$ is is not present. It follows that in the thermodynamic limit, contribute only the cases where $J$ is a set of paired momenta. Hence

$$\mathrm{pf}B(\boldsymbol{\kappa}) = \left[\sum_{\substack{J \subset \boldsymbol{\kappa} \\ J \text{ paired}}} \epsilon^{|J|/2}\, \mathrm{pf}B_J(\boldsymbol{\kappa})\right](1 + \mathcal{O}(L^{-1})). \tag{128}$$

We now observe that $B_J(\boldsymbol{\kappa})$ is exactly $D_t(\boldsymbol{\mu}) - D_t(\boldsymbol{\mu})^T$ with $|\sigma(\boldsymbol{\mu})| = |J|$, in the limit $\boldsymbol{\mu} \to \boldsymbol{\kappa}$ and $\sigma(\boldsymbol{\mu})$ becoming a set of paired momenta, equal to $J$. Hence in the thermodynamic limit $(-1)^{N/2}\mathrm{pf}B(\boldsymbol{\kappa})$ is the left-hand side of (125) multiplied by $\epsilon^{|\sigma(\boldsymbol{\mu})|/2}$, summed over $|\sigma(\boldsymbol{\mu})|$ and $\sigma(\boldsymbol{\mu})$, in the limit of a set of paired momenta equal to $\boldsymbol{\kappa}$.

Now, using (122) we observe that in the thermodynamic limit

$$\begin{aligned}
(-1)^{N/2}\mathrm{pf}B(\boldsymbol{\kappa}) &= \mathrm{Pf}[\mathrm{Jd} + (1+\epsilon)\mathcal{D}_{\rho, t}]\prod_{\kappa \in \pi(\boldsymbol{\kappa})} h_t(\kappa) \\
&= \mathrm{Pf}[\mathrm{Jd} + \mathcal{D}_{\rho, t}]\mathrm{Pf}[\mathrm{Jd} + \epsilon\tilde{\mathcal{F}}_{\rho, t}]\prod_{\kappa \in \pi(\boldsymbol{\kappa})} h_t(\kappa),
\end{aligned} \tag{129}$$

with $\tilde{\mathcal{F}}_{\rho, t}(\lambda, \mu) = \frac{\mathcal{F}_{\rho, t}(\lambda, \mu)}{h_t^+(\lambda)h_t^+(\mu)}$. Using (64) to expand $\mathrm{Pf}[\mathrm{Jd}+\epsilon\tilde{\mathcal{F}}_{\rho, t}]$, we obtain that in the thermodynamic limit, $(-1)^{N/2}\mathrm{pf}B(\boldsymbol{\kappa})$ is the right-hand side of (125) multiplied by $\epsilon^{|\sigma(\boldsymbol{\mu})|/2}$, summed over $|\sigma(\boldsymbol{\mu})|$ and $\sigma(\boldsymbol{\mu})$, in the limit of a set of paired momenta equal to $\boldsymbol{\kappa}$, with $\mathfrak{s}(\sigma(\boldsymbol{\mu}), t) = 1$. Thus by comparing the order $\epsilon^{|\sigma(\boldsymbol{\mu})|/2}$ of the two expansions in $\epsilon$ (128) and (129) we deduce $\mathfrak{s}(\sigma(\boldsymbol{\mu}), t) = 1$ for all $|\sigma(\boldsymbol{\mu})|$. Hence we have in the thermodynamic limit

$$(-1)^{N/2}\mathrm{pf}[D_t(\boldsymbol{\mu}) - D_t(\boldsymbol{\mu})^T] = \frac{1}{L^{|\sigma(\boldsymbol{\mu})|/2}}\mathrm{Pf}[\mathrm{Jd} + \mathcal{D}_{\rho, t}] \tag{130}$$

$$\times \mathrm{pf}\left[\mathcal{F}_{\rho, t}(\nu_i, \nu_j)\right]_{\nu_i, \nu_j \in \sigma(\boldsymbol{\mu})}\prod_{\mu \in \pi(\boldsymbol{\mu})} h_t(\mu). \tag{131}$$

### 3.6.3 Summation over the $e^{-itH(h,\gamma)}$ form factors

Returning to (112) we now see that the form factor of $e^{-itH(h,\gamma)}$ given in Lemma 4 imposes that $\sigma(\boldsymbol{\lambda}) = \sigma(\boldsymbol{\mu})$. This permits us to write

$$\begin{aligned}
C^{xx}(\ell, t) = A_t^{\mathrm{R}}A_{-t}^{\mathrm{R}*}\sum_{\substack{\boldsymbol{\nu} \subset \mathrm{NS} \\ \boldsymbol{\nu} \cap (-\boldsymbol{\nu}) = \emptyset}}\frac{\prod_{\nu \in \boldsymbol{\nu}} e^{i\ell\nu}}{L^{|\boldsymbol{\nu}|}}\sum_{\substack{\boldsymbol{\lambda}, \boldsymbol{\mu} \subset \\ \mathrm{NS}_+ - \{\boldsymbol{\nu}, -\boldsymbol{\nu}\}}}{}^{\mathrm{NS}}_{01}\langle\bar{\boldsymbol{\lambda}} \cup \boldsymbol{\nu}|e^{-iH(h,\gamma)t}|\bar{\boldsymbol{\mu}} \cup \boldsymbol{\nu}\rangle^{\mathrm{NS}}_{01} \\
\times \prod_{\lambda \in \boldsymbol{\lambda}} h_{-t}^*(\lambda)\prod_{\mu \in \boldsymbol{\mu}} h_t(\mu)\, \mathrm{Pf}[\mathrm{Jd} + \mathcal{D}_{\rho, t}]\, \mathrm{Pf}[\mathrm{Jd} + \mathcal{D}'_{\rho', -t}] \\
\times \mathrm{pf}\left[\mathcal{F}_{\rho, t}(\nu_i, \nu_j)\right]_{\nu_i, \nu_j \in \boldsymbol{\nu}}\mathrm{pf}\left[\mathcal{F}'_{\rho', -t}(\nu_i, \nu_j)\right]_{\nu_i, \nu_j \in \boldsymbol{\nu}},
\end{aligned} \tag{132}$$

where $\rho$ and $\rho'$ are the root densities corresponding to $\boldsymbol{\mu}$ and $\boldsymbol{\lambda}$ respectively. At fixed $\boldsymbol{\nu}$, given the form factor of $e^{-itH(h,\gamma)}$ in Lemma 4, the summand is of the form of Lemma 8 with

$$f = iKh_{-t}^* \frac{1-e^{-2it\varepsilon}}{1+K^2 e^{-2it\varepsilon}}, \ g = -iKh_t \frac{1-e^{-2it\varepsilon}}{1+K^2 e^{-2it\varepsilon}}, \ h = \frac{(1+e^{-2it\varepsilon}K^2)(1+e^{-2it\varepsilon}/K^2)}{(1-e^{-2it\varepsilon})^2}, \quad (133)$$

and with $\mathrm{NS}_+$ replaced by $\mathrm{NS}_+ - \{\boldsymbol{\nu} \cup -\boldsymbol{\nu}\}$. To apply Lemma 8, let us first investigate the denominator of Eq (194). We note that the form factor of $e^{-iH(h,\gamma)t}$ in Lemma 4 generates a factor

$$e^{-it\mathfrak{e}^{\mathrm{NS}}} \prod_{k \in \mathrm{NS}_+} \frac{1+K^2(k)e^{-2it\varepsilon(k)}}{1+K^2(k)}. \quad (134)$$

Moreover, from (111) we have

$$A_t^{\mathrm{NS}} A_{-t}^{\mathrm{NS}*} e^{-it\mathfrak{e}^{\mathrm{NS}}} \prod_{k \in \mathrm{NS}_+} \frac{1+K^2(k)e^{-2it\varepsilon(k)}}{1+K^2(k)}$$

$$= \prod_{k \in \mathrm{NS}_+} \frac{1+K^2(k)e^{-2it\varepsilon(k)}}{[1 + h_t h_{-t}^* K^2 + (h_t h_{-t}^* + K^2)e^{-2it\varepsilon} - iK(h_t - h_{-t}^*)(1-e^{-2it\varepsilon})](k)}, \quad (135)$$

which is precisely the inverse of the denominator in (194). Hence, defining the following (complex) root density

$$\rho_t = \frac{1}{2\pi} \frac{-iK(1-e^{-2it\varepsilon})h_t + (K^2+e^{-2it\varepsilon})h_t h_{-t}^*}{1+h_t h_{-t}^* K^2 + (h_t h_{-t}^* + K^2)e^{-2it\varepsilon} - iK(h_t - h_{-t}^*)(1-e^{-2it\varepsilon})}, \quad (136)$$
$$\rho_t' = [\rho_{-t}]^*,$$

appearing in Lemma 8, we obtain

$$C^{xx}(\ell,t) = \phi_\infty(t)\phi_\infty(-t)^* \sum_{\substack{\boldsymbol{\nu} \subset \mathrm{NS} \\ \boldsymbol{\nu} \cap (-\boldsymbol{\nu}) = \emptyset}} \frac{\prod_{\nu \in \boldsymbol{\nu}} s_{\ell,t}(\nu)}{L^{|\boldsymbol{\nu}|}} \, \mathrm{Pf}[\mathrm{Jd} + \mathcal{D}_{\rho_t,t}] \mathrm{Pf}[\mathrm{Jd} + \mathcal{D}'_{\rho_t',-t}]$$

$$\times \ \mathrm{pf}\left[\mathcal{F}_{\rho_t,t}(\nu_i,\nu_j)\right]_{\nu_i,\nu_j \in \boldsymbol{\nu}} \mathrm{pf}\left[\mathcal{F}'_{\rho_t',-t}(\nu_i,\nu_j)\right]_{\nu_i,\nu_j \in \boldsymbol{\nu}}, \quad (137)$$

where

$$s_{\ell,t}(z) = \frac{1}{2\pi} \frac{[1+K^2(z)]e^{i(\ell z - t\varepsilon(z))}}{[1+h_t h_{-t}^* K^2 + (h_t h_{-t}^* + K^2)e^{-2it\varepsilon} - iK(h_t - h_{-t}^*)(1-e^{-2it\varepsilon})](z)}, \quad (138)$$

and

$$\phi_\infty(t) = \lim_{L \to \infty} \frac{A_t^{\mathrm{R}}}{A_t^{\mathrm{NS}}}. \quad (139)$$

The factor $s_{\ell,t}(z)$ arises from the terms in (182) corresponding to the unpaired momenta $\boldsymbol{\nu}$, and the fact that Lemma 8 is applied with $\mathrm{NS}_+$ replaced by $\mathrm{NS}_+ - \{\boldsymbol{\nu} \cup -\boldsymbol{\nu}\}$. The phase $\phi_\infty(t)$ is identical to the phase discussed in Section 2.4.2. However, since here the operators involved in the dynamical correlation are time-evolved with a Hamiltonian with a constant magnetic field and anisotropy, the phase can be expressed only in terms of quantities at $t$. It can be straightforwardly computed with Eqs (111) and (110), and so is much easier to evaluate numerically than the generic phase discussed in Section 2.4.2.

### 3.6.4 Representation as a product of Pfaffians

In the thermodynamic limit the sums over the unpaired momenta $v \in \boldsymbol{v}$ in (137) can be converted into $|\boldsymbol{v}| = 2n$-fold integrals over $[-\pi, \pi]$, because the cases where $v_i = -v_j$ that are excluded in (137) are negligible by at least a factor of $L$. This provides us with a multiple-integral representation of the form

$$
C^{xx}(\ell, t) = \phi_\infty(t)\phi_\infty(-t)^* \operatorname{Pf}[\mathbf{Jd} + \mathcal{D}_{\rho_t, t}] \operatorname{Pf}[\mathbf{Jd} + \mathcal{D}'_{\rho^*_{-t}, -t}]
$$
$$
\times \sum_{n \ge 0} \frac{1}{(2n)!} \int_{-\pi}^{\pi} \cdots \int_{-\pi}^{\pi} \prod_{j=1}^{2n} s_{\ell, t}(z_j) \operatorname*{pf}_{i,j}[\mathcal{F}_{\rho_t, t}(z_i, z_j)] \operatorname*{pf}_{i,j}[\mathcal{F}'_{\rho^*_{-t}, -t}(z_i, z_j)] dz_1 \dots dz_{2n},
$$

(140)

with the convention that the term for $n = 0$ in the series is equal to 1. We now observe that

$$
\operatorname*{pf}_{i,j}[\mathcal{F}_{\rho_t, t}(z_i, z_j)] \operatorname*{pf}_{i,j}[\mathcal{F}'_{\rho^*_{-t}, -t}(z_i, z_j)] \prod_{j=1}^{2n} s_{\ell, t}(z_j)
$$
$$
= \operatorname{pf}\begin{bmatrix} (s_{\ell, t}(z_i)s_{\ell, t}(z_j)\mathcal{F}_{\rho_t, t}(z_i, z_j))_{1 \le i, j \le 2n} & 0 \\ 0 & (\mathcal{F}'_{\rho^*_{-t}, -t}(z_i, z_j))_{1 \le i, j \le 2n} \end{bmatrix}
$$
$$
= \operatorname*{pf}_{1 \le i, j \le 2n}\begin{bmatrix} s_{\ell, t}(z_i)s_{\ell, t}(z_j)\mathcal{F}_{\rho_t, t}(z_i, z_j) & 0 \\ 0 & \mathcal{F}'_{\rho^*_{-t}, -t}(z_j, z_i) \end{bmatrix}.
$$

(141)

The swap of the arguments of $\mathcal{F}'_{\rho^*_{-t}, -t}$ in the last line compensates the sign factor $(-1)^n$ that results from changing the $2n \times 2n$ block $2 \times 2$ matrix into a $2 \times 2$ block $2n \times 2n$ matrix. Eqn. (141) allows us to recast the series over $n$ of (140) in the form of the block Fredholm Pfaffian (130) with $n$ restricted to be even, namely

$$
\sum_{n \ge 0} \frac{1}{(2n)!} \int_{-\pi}^{\pi} \cdots \int_{-\pi}^{\pi} \prod_{j=1}^{2n} s_{\ell, t}(z_j) \operatorname*{pf}_{i,j}[\mathcal{F}_{\rho_t, t}(z_i, z_j)] \operatorname*{pf}_{i,j}[\mathcal{F}'_{\rho^*_{-t}, -t}(z_i, z_j)] dz_1 \dots dz_{2n}
$$
$$
= \frac{1}{2}\left(\operatorname{Pf}[\mathbf{Jd} + \boldsymbol{F}_{\ell, t}] + \operatorname{Pf}[\mathbf{Jd} - \boldsymbol{F}_{\ell, t}]\right),
$$

(142)

where $\boldsymbol{F}_{\ell, t}(x, y)$ is a $2 \times 2$ matrix-valued function on $[-\pi, \pi] \times [-\pi, \pi]$

$$
\boldsymbol{F}_{\ell, t}(x, y) = \begin{pmatrix} s_{\ell, t}(x)s_{\ell, t}(y)\mathcal{F}_{\rho_t, t}(x, y) & 0 \\ 0 & \mathcal{F}'_{\rho^*_{-t}, -t}(y, x) \end{pmatrix}.
$$

(143)

From the fact that the $2 \times 2$ kernel $\boldsymbol{F}_{\ell, t}$ is diagonal, using that the Pfaffian is multiplied by $-1$ whenever a row and the corresponding columns are multiplied simultaneously by $-1$, we find

$$
\operatorname{Pf}[\mathbf{Jd} - \boldsymbol{F}_{\ell, t}] = \operatorname{Pf}[\mathbf{Jd} + \boldsymbol{F}_{\ell, t}].
$$

(144)

Hence putting everything together we arrive at our final result

$$
\boxed{C^{xx}(\ell, t) = \phi_\infty(t)\phi_\infty(-t)^* \quad \operatorname{Pf}[\mathbf{Jd} + \mathcal{D}_{\rho_t, t}]\operatorname{Pf}[\mathbf{Jd} + \mathcal{D}'_{\rho^*_{-t}, -t}] \operatorname{Pf}[\mathbf{Jd} + \boldsymbol{F}_{\ell, t}].}
$$

(145)

Eq (145) again assumes that the functions $h_t$ and $h_{-t}$ are regular.

A remark on how generalizable this calculation is.

# 4 Applications

In this section we apply the results reported above in a number of settings. For simplicity we focus on the case of the transverse-field Ising chain $\gamma = 1$.

## 4.1 Order parameter and the Kibble-Zurek mechanism

As a first application we consider the time-dependence of the order parameter in the transverse-field Ising model for a ramp of the magnetic field through the critical point. While this and closely related non-equilibrium protocols have been previously studied in great detail [71, 88, 94, 98, 103, 107, 108] in connection to the Kibble-Zurek mechanism [102, 109–112], we are not aware of any results on the dynamics of the order parameter in the thermodynamic limit. We will consider time-dependent magnetic fields

$$
\begin{aligned}
&\text{(i)} \qquad h(t) = h_0 + \alpha t, \qquad \gamma = 1, \\
&\text{(ii)} \qquad h(t) = 1 + \frac{(\alpha t - 1 + h_0)^3}{(1 - h_0)^2}, \qquad \gamma = 1,
\end{aligned}
\tag{146}
$$

that cross the critical point linearly (case (i)) or cubically (case (ii)) with a speed parameter $\alpha$, and assume that the system is initialized in the ground state for $0 < h_0 < 1$ at time $t = 0$. The presence of spontaneous symmetry breaking in the thermodynamic limit can be accounted for by working with the following initial state, *cf.* Refs [76, 77]

$$
|\psi_{h_0}(0)\rangle = \frac{|0\rangle_{h_0 1}^{\mathrm{R}} + \alpha_{01;0}^\dagger |0\rangle_{h_0 1}^{\mathrm{NS}}}{\sqrt{2}}.
\tag{147}
$$

The time evolution of the order parameter in the thermodynamic limit is then obtained from (80), (52)

$$
\langle \psi_{h_0}(t) | \sigma_\ell^x | \psi_{h_0}(t) \rangle = \Re \left( \phi_\infty(t) \mathrm{Det}[\mathrm{Id} + \mathcal{M}[\rho_s]] \right).
\tag{148}
$$

Here $\mathcal{M}$ is given in (81) with

$$
h(k) = \frac{i K_{01;h(t)1}(k) + f_t(k)}{1 + i K_{01;h(t)1}(k) f_t(k)},
\tag{149}
$$

and $f_t(k)$ is the solution of the nonlinear differential equation (45). Importantly the phase factor $\phi_\infty(t)$ is not always equal to one in this case. We find that there is a sequence of times $t_n^*$ and associated magnetic fields $h_n^* = h(t_n^*)$ such that

$$
\phi_\infty(t) = (-1)^n, \qquad t_n < t < t_{n+1}, n = 0, 1, \dots.
\tag{150}
$$

As an example we consider $h_0 = 0.6$ and $\alpha = 0.3$ for the linear ramp of case (i). Then

$$
\begin{aligned}
t_0^* &\approx 2.883547, \quad h_0^* \approx 1.465064, \\
t_1^* &\approx 4.591509, \quad h_1^* \approx 1.977452.
\end{aligned}
\tag{151}
$$

We observe, in agreement with the discussion of Section 2.4.2, that this behaviour arises from the vanishing of the overlaps $\langle \psi_{h_0}(t) | \psi_0(0) \rangle$ at particular times $t_n^*$ in the thermodynamic limit, which is equivalent to the function $f_t(k)$ becoming singular at times $t_n^*$ for particular wave numbers $k_n^*$. We note however that the Fredholm determinant appearing in (148) also exhibits discontinuities at times $t_n^*$ in such a way that the resulting magnetization $\langle \sigma^x(t) \rangle$ is a continuous function of time, as expected on physical grounds.

The results of a numerical evaluation of the Fredholm determinant expression for the order parameter during the quench are shown in Fig. 3. We use (59) and [113] to compute the determinant and use a quadrature rule with up to 4000 points. We stress that by construction we are considering the magnetization per site *in the thermodynamic limit*.

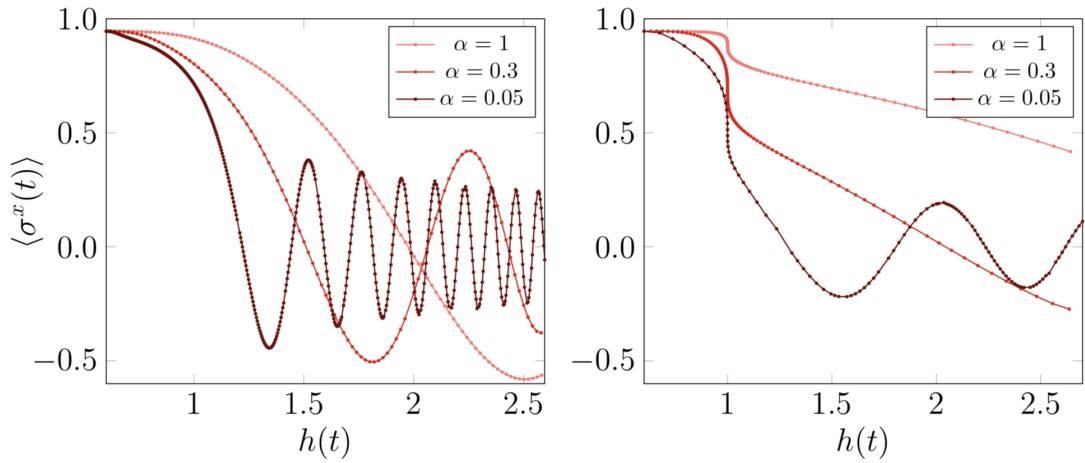

Figure 3: Order parameter expectation value $\langle \sigma^x(t) \rangle$ as a function of $h(t)$, with a ramp crossing the critical point linearly $h(t) = 0.6 + \alpha t$ (left) and cubically $h(t) = 1 + 6.25 \times (t\alpha - 0.4)^3$ (right).

We first consider linear ramps across the quantum criticial point starting in the ordered phase, i.e. case (i) in (146) with $h_0$=0.6. We see that ramping up the magnetic field initially leads to a reduction of $\langle \sigma^x_j(t) \rangle$, the size of which depends on the ramp rate $\alpha$. For very fast $\alpha$, $\langle \sigma^x_j(t) \rangle$ is expected to remain essentially pinned to its value at $t = 0$: this corresponds to a sudden approximation and is closely related to the situation encountered in a quantum quench. A slower ramp rate is expected to result in a faster reduction of $\langle \sigma^x_j(t) \rangle$ at early times. Both of these expectations are borne out by the numerical results shown in Fig. 3. At later times the magnetization per site displays an oscillatory behaviour. In the scaling regime around the critical field $h = 1$ this behaviour has been analyzed in some detail in Ref. [94]. For a very slow ramp rate the magnetization closely follows the magnetic field dependence in the ground state, as expected by the adiabatic approximation, until $h \approx 1$, where adiabadicity breaks down and Kibble-Zurek physics ensues. We next consider a nonlinear ramp starting at the same initial field $h(0) = 0.6$ and whose derivative vanishes at the critical point, given by case (ii) in (146). On a very qualitative level the time dependence of the order parameter is similar to the linear ramp case in that oscillations ensue after an initial decay.

## 4.2 Order parameter in periodically driven systems

The results derived in the previous sections allow for a systematic study of the thermodynamic limit of Floquet physics where the driving magnetic field and anisotropy are periodic functions of time, see e.g. Refs [114–122]. In Fig. 4 we show the order parameter $\langle \sigma^x_\ell(t) \rangle$ of a system initialized in the ground state at $h = 0$ and then driven periodically with frequency $\omega$

$$h(t) = \frac{1 - \cos(\pi \omega t)}{2}, \qquad \gamma = 1. \tag{152}$$

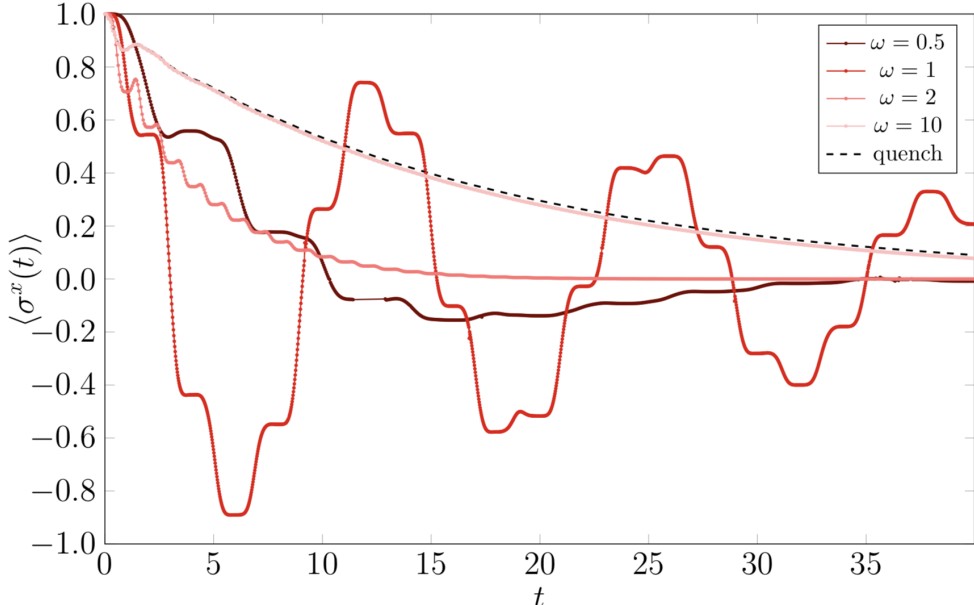

Figure 4: Order parameter expectation value $\langle \sigma^x(t) \rangle$ as a function of $t$, with the variation of magnetic field $h(t) = \frac{1}{2}(1 - \cos \pi \omega t)$. In dashed is indicated the time-evolution after a sudden quench $h(t) = \frac{1}{2}$ for $t > 0$.

At large frequencies $\omega$ we expect to recover the results for evolution with the time-averaged Hamiltonian [118], which corresponds to a quantum quench where the system in initialized in the ground state of $H(0, 1)$ and then time-evolved with $H(\frac{1}{2}, 1)$. We see that the time evolution for $\omega = 10$ is indeed very close to this limit. At late times the system synchronizes and can be described by a "periodic generalized Gibbs ensemble" [117]. In particular this implies that the order parameter should vanish, which is indeed what we observe in Fig. 4. In the limit of low frequencies $\omega \approx 0$ the behaviour is initially adiabatic and the order parameter follows the ground state value at the corresponding magnetic field $h(t)$. As $t \to \omega^{-1}$ the magnetic field $h(t)$ approaches its critical value and adiabaticity breaks down and Kibble-Zurek physics ensues. For frequencies $\omega > 2$ the magnetization is seen to decay towards zero with only weak oscillations on top of the decay. For frequencies $\omega \approx 1$ there are strong oscillations that decay in time. Interestingly, for lower frequencies the oscillatory behaviour becomes less pronounced.

### 4.3  Dynamical correlations after a sudden quench

In this section we illustrate formula (145) for dynamical correlations in a CE for a particular case of the general scenario discussed in section 2.4. We initialize the state of the system $|\psi(0)\rangle$ in the ground state of the transverse field Ising model with magnetic field $h_0$, i.e. $H(h_0, 1)$, and suddenly change the magnetic field to $h$, triggering a non-trivial time evolution of the state $|\psi(t)\rangle$. We are then interested in the connected non-equal time order parameter correlation function

$$\langle \sigma^x_{\ell+1}(t_1) \sigma^x_1(t_2) \rangle_c = \langle \psi(t_2) | \sigma^x_{\ell+1} e^{i(t_1-t_2)H(h,\gamma)} \sigma^x_1 | \psi(t_1) \rangle - \langle \sigma^x_{\ell+1}(t_1) \rangle \langle \sigma^x_1(t_2) \rangle . \qquad (153)$$

Numerical results for $\langle \sigma^x_{\ell+1}(t_1) \sigma^x_1(t_2) \rangle_c$ for a quench from $h_0 = 0.1$ to $h = 0.9$, obtained by numerically evaluating (145), are shown in Fig. 5. One sees that there are several regions where the 2-point function is negligibly small. These light-cone structures can be understood with

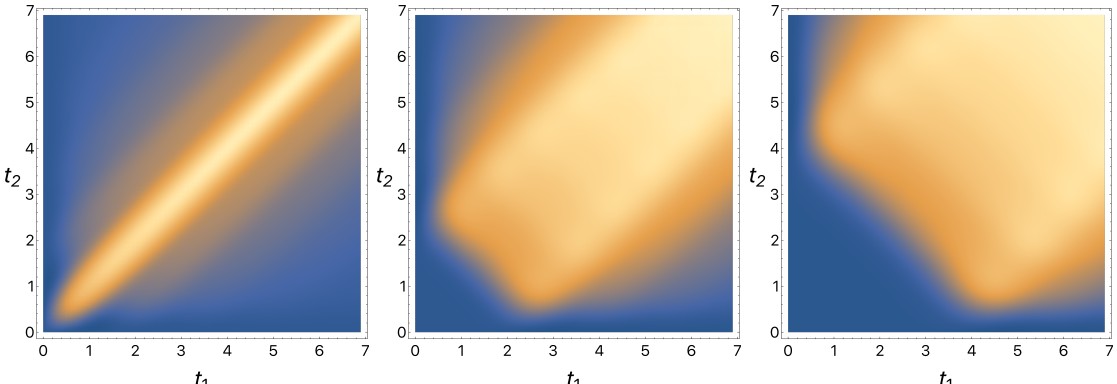

Figure 5: Density plots of the connected non-equal time correlation $\langle \sigma^x_{\ell+1}(t_2)\sigma^x_1(t_1)\rangle_c$ after a quantum quench as a function of $t_1$ and $t_2$, for $\ell = 0, 3, 6$ from left to right. The three plots use different color scales.

the quasi-particle picture initially proposed to describe the growth of entanglement entropy after a quench [83, 123]. According to this picture, the effect of the quench is to create pairs of quasi-particles at each position in the chain that evolve with velocities $\pm v(k) = \pm\partial_k \varepsilon_{h\gamma}(k)$, and a non-zero connected correlation can occur between two points in space-time only if a quasi-particle can propagate from one to the other. Let us fix $t_1 < t_2$ and set $\delta = t_2 - t_1$. According to this quasi-particle picture, the operator $\sigma^x_{\ell+1}(t_1)$ can be considered as a local operator with support in $[\ell + 1 - v_{\max}t_1, \ell + 1 + v_{\max}t_1]$, with $v_{\max} = \max_k |v(k)|$. The condition for the connected correlator to be non-negligible is for the supports of $\sigma^x_{\ell+1}(t_1)$ and $\sigma^x_1(t_2)$ (or equivalently their backward light-cones) to overlap. This explains why for $t_1 + t_2 < \frac{\ell}{v_{\max}}$ the connected 2-point function is negligibly small. This corresponds to the triangular blue regions in the bottom left corners of Figs 5, which grow with $\ell$. On the other hand, we expect on physical grounds that the effects of making a local perturbation at time $t_1$ will become increasingly difficult to detect if we wait long enough and connected correlations should therefore decay with respect to the time difference $|t_2 - t_1|$ when the latter gets large. This explains the smallness of the connected two-point functions observed in the upper left and bottom right corners of 5.

## 5  Discussion and outlook

In this work we have addressed the problem of computing out-of-equilibrium observables in the XY spin chain subject to arbitrary time variations of the magnetic field $h(t)$ and anisotropy $\gamma(t)$. We obtained closed-form expressions for the thermodynamic limit of the order parameter expectation value, dynamical two-point function and static three-point function, as well as of the full counting statistics of the transverse magnetization. These expressions are valid for all times and for arbitrary distances. They hold not only for out-of-equilibrium situations, but also in the wider context of the Coherent Ensemble as introduced in the text. We emphasize that to the best of our knowledge no exact explicit expressions in the thermodynamic limit were known for the expectation value of operators that are non-local in the underlying fermions, namely for the order parameter one-point function, two-point dynamical correlation and three-point static correlation. While the expectation values of operators that are local in the underlying fermions can be straightforwardly computed using Wick's theorem and do not require the Fredholm determinant expressions derived in this paper, our method provides a

unified approach based on form factor summation, hence better suited for generalization to the interacting case where no Wick's theorem holds. Despite the free nature of the XY model, the problem of performing the spectral sum over form factors was previously solved only for free models with $U(1)$ symmetry such as the impenetrable Bose gas or the XX chain.

In our derivation of these results we have followed a different route than the ones traditionally used in the computation of out-of-equilibrium dynamics in integrable models. Our approach relies on remarkable properties of *coherent states*, that are weighted superpositions (in a precise manner) of exponentially many eigenstates of the Hamiltonian, that in a sense behave more smoothly than pure eigenstates and are easier to manipulate. Crucially they stay coherent when expressed in terms of the eigenstates of the Hamiltonian at other values of $h$ and $\gamma$, which allows one to carry out the calculations in a preferred simple basis, such as $h = 0$ and $\gamma = 1$, where the form factors are exactly Cauchy determinants. Efficient summation formulas exploiting both the coherent state structure and the form factor determinant structure eventually lead to our results.

Our work opens up a number of future directions. The first direction is to determine the asymptotic behaviour of the various correlation functions considered here. A second direction is to investigate whether some ideas of this fruitful approach can be generalized to an interacting case. Although the coherent state structure used in this paper is rather fragile, there could be analogous macroscopic superpositions of eigenstates in an interacting model that enjoy similar interesting properties.

**Acknowledgements.** H.D. acknowledges support from the European Research Council under the European Union Horizon 2020 Research and Innovation Programme via Grant Agreement No. 804213-TMCS. E.G. acknowledges support from the EPSRC under grant EP/S020527/1.

# A  Diagonalization of the XY model in a field

## A.1  Mapping to free fermions

In this appendix we review how to diagonalize the XY Hamiltonian with magnetic field $h$ and anisotropy $\gamma$

$$H(h,\gamma) = -\sum_{j=1}^{L} \frac{1+\gamma}{2}\sigma_j^x \sigma_{j+1}^x + \frac{1-\gamma}{2}\sigma_j^y \sigma_{j+1}^y + h\sigma_j^z \,, \tag{154}$$

where $\sigma_j^\alpha$ are the Pauli matrices at site $j$ and

$$\sigma_{L+1}^\alpha = \sigma_1^\alpha\,, \quad \alpha = x, y, z. \tag{155}$$

The quantum XY chain is mapped to a model of spinless fermions by means of a Jordan-Wigner transformation. Defining $\sigma_j^\pm = \left(\sigma_j^x \pm i\sigma_j^y\right)/2$ we construct spinless fermion creation and annihilation operators by

$$c_l^\dagger = \prod_{j=1}^{l-1} \sigma_j^z \sigma_l^-\,, \quad \{c_j, c_l^\dagger\} = \delta_{j,l}. \tag{156}$$

The inverse transformation is

$$\sigma_j^z = 1 - 2c_j^\dagger c_j\,, \quad \sigma_j^x = \prod_{l=1}^{j-1}(1 - 2c_l^\dagger c_l)(c_j + c_j^\dagger)\,, \quad \sigma_j^y = i\prod_{l=1}^{j-1}(1 - 2c_l^\dagger c_l)(c_j^\dagger - c_j). \tag{157}$$

The Hamiltonian can be expressed in terms of the fermions as

$$H(h,\gamma) = -\sum_{j=1}^{L-1}\frac{1+\gamma}{2}\big[c_j^\dagger - c_j\big]\big[c_{j+1} + c_{j+1}^\dagger\big] - \sum_{j=1}^{L-1}\frac{1-\gamma}{2}\big[c_j^\dagger + c_j\big]\big[c_{j+1} - c_{j+1}^\dagger\big]$$

$$-h\sum_{j=1}^{L}[1-2c_j^\dagger c_j]$$

$$- e^{i\pi\hat{N}}\frac{1+\gamma}{2}(c_L - c_L^\dagger)(c_1 + c_1^\dagger) - e^{i\pi\hat{N}}\frac{1-\gamma}{2}(c_L + c_L^\dagger)(c_1^\dagger - c_1), \tag{158}$$

where

$$\hat{N} = \sum_{j=1}^{L} c_j^\dagger c_j. \tag{159}$$

As $[H, e^{i\pi\hat{N}}] = 0$ we may diagonalize the two operators simultaneously. The Hamiltonian is thus block diagonal $H = H^{\mathrm{NS}} \oplus H^{\mathrm{R}}$, where $H^{\mathrm{NS,R}}$ act on the subspaces of the Fock space with an even/odd number of fermions respectively.

## A.2 Even fermion number

In the sector with an even number of fermions we have $e^{i\pi\hat{N}} = 1$ and the Hamiltonian can be written in the form

$$H^{\mathrm{NS}}(h,\gamma) = -\sum_{j=1}^{L}\frac{1+\gamma}{2}\big[c_j^\dagger - c_j\big]\big[c_{j+1} + c_{j+1}^\dagger\big] - \sum_{j=1}^{L}\frac{1-\gamma}{2}\big[c_j^\dagger + c_j\big]\big[c_{j+1} - c_{j+1}^\dagger\big]$$

$$-h\sum_{j=1}^{L}[1-2c_j^\dagger c_j], \tag{160}$$

where we have imposed antiperiodic boundary conditions on the fermions

$$c_{L+1} = -c_1. \tag{161}$$

The Hamiltonian $H^{\mathrm{NS}}$ is diagonalized by going to Fourier space

$$c(k_n) = \frac{1}{\sqrt{L}}\sum_{j=1}^{L} c_j\, e^{ik_n j}, \tag{162}$$

where $k_n$ are quantized according to (161)

$$k_n = \frac{2\pi(n+1/2)}{L}, \quad n = -\frac{L}{2}, \dots \frac{L}{2} - 1. \tag{163}$$

The antiperiodic sector is commonly referred to as Neveu-Schwarz (NS) sector. Introducing Bogoliubov fermions by

$$c(k_n) = \cos(\theta_{k_n}/2)\alpha_{h\gamma;k_n} + i\sin(\theta_{k_n}/2)\alpha_{h\gamma;-k_n}^\dagger,$$

$$c^\dagger(-k_n) = i\sin(\theta_{k_n}/2)\alpha_{h\gamma;k_n} + \cos(\theta_{k_n}/2)\alpha_{h\gamma;-k_n}^\dagger, \tag{164}$$

where the Bogoliubov angle fulfils

$$\tan\theta_k = \left[\frac{\gamma\sin(k)}{\cos(k)-h}\right], \tag{165}$$

the Hamiltonian becomes diagonal

$$H^{\mathrm{NS}}(h,\gamma) = \sum_{k \in \mathrm{NS}} \varepsilon_{h\gamma}(k) \left[ \alpha^\dagger_{h\gamma;k} \alpha_{h\gamma;k} - \frac{1}{2} \right]. \tag{166}$$

Here the dispersion relation is

$$\varepsilon_{h\gamma}(k) = 2\sqrt{(h - \cos k)^2 + \gamma^2 \sin^2 k}. \tag{167}$$

A basis for the Fock space in the sector with even fermion number is then given by

$$|k_1, \ldots, k_{2m}\rangle^{\mathrm{NS}}_{h\gamma} = \prod_{j=1}^{2m} \alpha^\dagger_{h\gamma;k_j} |0\rangle^{\mathrm{NS}}_{h\gamma}, \quad k_j \in \mathrm{NS}, \tag{168}$$

where the fermion vacuum $|0\rangle^{\mathrm{NS}}_{h\gamma}$ is the state annihilated by all $\alpha_{h\gamma;k_j}$ $(j = -\frac{L}{2}, \ldots, \frac{L}{2} - 1)$.

### A.3   Odd fermion number

In the sector with an odd number of fermions we have $e^{i\pi\hat{N}} = -1$. The Hamiltonian can again be written in the form

$$H^{\mathrm{R}}(h,\gamma) = -\sum_{j=1}^{L} \frac{1+\gamma}{2} \left[ c^\dagger_j - c_j \right] \left[ c_{j+1} + c^\dagger_{j+1} \right] - \sum_{j=1}^{L} \frac{1-\gamma}{2} \left[ c^\dagger_j + c_j \right] \left[ c_{j+1} - c^\dagger_{j+1} \right]$$
$$-h \sum_{j=1}^{L} [1 - 2c^\dagger_j c_j], \tag{169}$$

but now we have to impose periodic boundary conditions on the fermions

$$c_{L+1} = c_1. \tag{170}$$

In Fourier space we therefore now have

$$c(p_n) = \frac{1}{\sqrt{L}} \sum_{j=1}^{L} c_j \, e^{i p_n j}, \tag{171}$$

where $p_n$ are quantized according to (170)

$$p_n = \frac{2\pi n}{L}, \quad n = -\frac{L}{2}, \ldots \frac{L}{2} - 1. \tag{172}$$

The periodic sector is known as Ramond sector. Defining Bogoliubov fermions $\alpha_{p_n}$ for $p_n \neq 0$ by

$$c(p_n) = \cos(\theta_{p_n}/2)\alpha_{h\gamma;p_n} + i\sin(\theta_{p_n}/2)\alpha^\dagger_{h\gamma;-p_n},$$
$$c^\dagger(-p_n) = i\sin(\theta_{p_n}/2)\alpha_{h\gamma;p_n} + \cos(\theta_{p_n}/2)\alpha^\dagger_{h\gamma;-p_n}, \tag{173}$$

we can express the Hamiltonian as

$$H^{\mathrm{R}}(h,\gamma) = \sum_{\substack{p \in \mathrm{R} \\ p \neq 0}} \varepsilon_{h\gamma}(p) \left[ \alpha^\dagger_{h\gamma;p} \alpha_{h\gamma;p} - \frac{1}{2} \right] - 2(1-h) \left[ \alpha^\dagger_{h\gamma;0} \alpha_{h\gamma;0} - \frac{1}{2} \right]. \tag{174}$$

A basis of the subspace of the Fock space with odd fermion numbers is then given by

$$|p_1, \ldots, p_{2m+1}\rangle^{\mathrm{R}}_{h\gamma} = \prod_{j=1}^{2m+1} \alpha^\dagger_{h\gamma;p_j} |0\rangle^{\mathrm{R}}_{h\gamma}, \quad p_j \in \mathrm{R}, \tag{175}$$

where the fermion vacuum $|0\rangle^{\mathrm{R}}_{h\gamma}$ is the state annihilated by all $\alpha_{h\gamma;p_j}$ $(j = -\frac{L}{2}, \ldots, \frac{L}{2} - 1)$.

# B  Useful lemmas

## B.1  Overlap and form factors

**Lemma 1.** *Let $|\boldsymbol{k}\rangle_{h\gamma}^{\text{NS}}$ and $|\boldsymbol{q}\rangle_{\tilde{h}\tilde{\gamma}}^{\text{NS}}$ two eigenstates of the XY Hamiltonian at different magnetic fields $h, \tilde{h}$ and anisotropies $\gamma, \tilde{\gamma}$. We define $\pi(\boldsymbol{k})$ as the subset of strictly positive elements $k_n \in \boldsymbol{k}$ such that $-k_n \in \boldsymbol{k}$ as well, and $\sigma(\boldsymbol{k})$ the subset of unpaired momenta, i.e. $k_j \in \boldsymbol{k}$ but $-k_j \notin \boldsymbol{k}$. $\pi(\boldsymbol{q})$ and $\sigma(\boldsymbol{q})$ are defined analogously. Then the following formula for the overlap between the two states holds*

$$
{}_{\tilde{h}\tilde{\gamma}}^{\text{NS}}\langle \boldsymbol{q}|\boldsymbol{k}\rangle_{h\gamma}^{\text{NS}} = \mathbf{1}_{\sigma(\boldsymbol{q})=\sigma(\boldsymbol{k})} \frac{\prod_{p\in\sigma(\boldsymbol{k})} \frac{1}{\cos(\theta_k^{\tilde{h}\tilde{\gamma}}-\theta_k^{h\gamma})/2} \prod_{p\in\pi(\boldsymbol{q})\perp\pi(\boldsymbol{k})} iK_{\tilde{h}\tilde{\gamma};h\gamma}(p)}{\prod_{p\in\text{NS}_+} \sqrt{1+K_{\tilde{h}\tilde{\gamma};h\gamma}^2(p)}}, \tag{176}
$$

*where we defined $\pi(\boldsymbol{q}) \perp \pi(\boldsymbol{k}) = \pi(\boldsymbol{q}) \cup \pi(\boldsymbol{k}) - (\pi(\boldsymbol{q}) \cap \pi(\boldsymbol{k}))$.*

*Proof.* As shown in Appendix A, we have the following relation between Bogoliubov fermion operators at different values of magnetic fields and anisotropies

$$
\alpha_{h\gamma;k} = \cos \frac{\theta_k^{\tilde{h}\tilde{\gamma}}-\theta_k^{h\gamma}}{2} \alpha_{\tilde{h}\tilde{\gamma};k} + i \sin \frac{\theta_k^{\tilde{h}\tilde{\gamma}}-\theta_k^{h\gamma}}{2} \alpha_{\tilde{h}\tilde{\gamma};-k}^{\dagger}. \tag{177}
$$

From this, we deduce the relation between the vacuum states

$$
|0\rangle_{h\gamma}^{\text{NS}} = \prod_{p\in\text{NS}_+} \left[ \frac{1+iK_{\tilde{h}\tilde{\gamma};h\gamma}(p)\alpha_{\tilde{h}\tilde{\gamma};-p}^{\dagger}\alpha_{\tilde{h}\tilde{\gamma};p}^{\dagger}}{\sqrt{1+K_{\tilde{h}\tilde{\gamma};h\gamma}^2(p)}} \right] |0\rangle_{\tilde{h}\tilde{\gamma}}^{\text{NS}}. \tag{178}
$$

Indeed, the right-hand side is annihilated by all the $\alpha_{h\gamma,k}$. From these relations one deduces the overlap given in the Lemma. $\qquad\square$

**Lemma 2** (Form factor of $e^{i\theta \sum_{j=1}^{\ell} \sigma_j^z}$)**.** *If $\boldsymbol{\lambda}$ and $\boldsymbol{\mu}$ have the same number of elements the following determinant representation holds*

$$
{}_{\infty\gamma}^{\text{NS}}\langle \boldsymbol{\lambda}|e^{i\theta \sum_{j=1}^{\ell} \sigma_j^z}|\boldsymbol{\mu}\rangle_{\infty\gamma}^{\text{NS}} = e^{i\theta\ell} \det E(\boldsymbol{\lambda},\boldsymbol{\mu}),
$$

$$
E(\boldsymbol{\lambda},\boldsymbol{\mu})_{jk} = \begin{cases} \frac{e^{-2i\theta}-1}{L} e^{i(\lambda_j-\mu_k)} \frac{1-e^{i\ell(\lambda_j-\mu_k)}}{1-e^{i(\lambda_j-\mu_k)}} & \text{if } \lambda_j \neq \mu_k, \\ 1 + \frac{\ell}{L}(e^{-2i\theta}-1) & \text{if } \lambda_j = \mu_k. \end{cases} \tag{179}
$$

*If they have different numbers of elements the form factor vanishes.*

*Proof.* Since the operator conserves the number of particles, $\boldsymbol{\lambda}$ and $\boldsymbol{\mu}$ must have the same number of particles for the form factor not to vanish. Let us denote this number by $N$. Using that at $h = \infty$ the Bogoliubov fermions reduce to the Jordan-Wigner fermions we have

$$
\begin{aligned}
{}_{\infty\gamma}^{\text{NS}}\langle \boldsymbol{\lambda}|e^{i\theta \sum_{j=1}^{\ell} \sigma_j^z}|\boldsymbol{\mu}\rangle_{\infty\gamma}^{\text{NS}} &= \frac{1}{L^N} \sum_{j_1\dots j_N} \sum_{k_1\dots k_N} {}_{\infty\gamma}^{\text{NS}}\langle 0|c_{j_N}\dots c_{j_1} e^{i\theta \sum_{j=1}^{\ell} \sigma_j^z} c_{k_1}^{\dagger}\dots c_{k_N}^{\dagger}|0\rangle_{\infty\gamma}^{\text{NS}} \\
&\qquad\qquad \times e^{-ik_1\mu_1-\dots-ik_N\mu_N} e^{ij_1\lambda_1+\dots+ij_N\lambda_N} \\
&= \frac{e^{i\theta\ell}}{L^N} \sum_{\sigma\in\mathfrak{S}_N} (-1)^{\sigma} \sum_{j_1\dots j_N} e^{ij_1(\lambda_1-\mu_{\sigma(1)})+\dots+ij_N(\lambda_N-\mu_{\sigma(N)})} e^{-2i\theta \sum_{q=1}^{N} \mathbf{1}_{j_q\leq\ell}} \\
&= \frac{e^{i\theta\ell}}{L^N} \sum_{\sigma\in\mathfrak{S}_N} (-1)^{\sigma} \prod_{q=1}^{N} \sum_{j=1}^{L} e^{ij(\lambda_q-\mu_{\sigma(q)})} e^{-2i\theta \mathbf{1}_{j\leq\ell}} \\
&= e^{i\theta\ell} \det E(\boldsymbol{\lambda},\boldsymbol{\mu}). \tag{180}
\end{aligned}
$$

$\qquad\square$

**Lemma 3** (Form factors of $\sigma^x$). *The form factors of $\sigma_\ell^x$ between energy eigenstates at $h = 0$, $\gamma = 1$ have the following determinant representation*

$$
\begin{aligned}
{}_{01}^{\mathrm{R}}\langle\boldsymbol{\lambda}\cup\{0\}|\sigma_\ell^x|\boldsymbol{\mu}\rangle_{01}^{\mathrm{NS}} =&\, (-1)^{N(N+1)/2}\left(\frac{2}{L}\right)^N e^{\frac{i}{2}(\sum_{\lambda\in\boldsymbol{\lambda}}\lambda+\sum_{\mu\in\boldsymbol{\mu}}\mu)}e^{-i\ell(\sum_{\lambda\in\boldsymbol{\lambda}}\lambda-\sum_{\mu\in\boldsymbol{\mu}}\mu)} \\
&\times \det\left[\frac{1}{e^{i\lambda_j}-e^{i\mu_k}}\right]_{jk}.
\end{aligned}
\tag{181}
$$

*Here $N$ is the number of elements in $\boldsymbol{\lambda}$ and $\boldsymbol{\mu}$. If they have different numbers of elements the form factor vanishes.*

*Proof.* See Refs [124–127]. $\qquad\square$

**Lemma 4** (Form factors of $e^{-itH(h,\gamma)}$). *The form factors of $e^{-itH}$ between energy eigenstates at $h = 0$, $\gamma = 1$ have the following representation*

$$
\begin{aligned}
{}_{01}^{\mathrm{NS}}\langle\boldsymbol{\lambda}|e^{-itH(h,\gamma)}|\boldsymbol{\mu}\rangle_{01}^{\mathrm{NS}} =&\, \mathbf{1}_{\sigma(\boldsymbol{\lambda})=\sigma(\boldsymbol{\mu})}e^{-it\mathfrak{E}^{\mathrm{NS}}}\prod_{k\in\mathrm{NS}_+}\frac{1+K^2(k)e^{-2it\varepsilon(k)}}{1+K^2(k)} \\
&\times \prod_{v\in\sigma(\boldsymbol{\lambda})}e^{-it\varepsilon(v)}\frac{1+K^2(v)}{1+e^{-2it\varepsilon(v)}K^2(v)} \\
&\times \prod_{\lambda\in\pi(\boldsymbol{\lambda})}iK(\lambda)\frac{1-e^{-2it\varepsilon(\lambda)}}{1+K^2(\lambda)e^{-2it\varepsilon(\lambda)}}\prod_{\mu\in\pi(\boldsymbol{\mu})}(-iK(\mu))\frac{1-e^{-2it\varepsilon(\mu)}}{1+K^2(\mu)e^{-2it\varepsilon(\mu)}} \\
&\times \prod_{v\in\pi(\boldsymbol{\lambda})\cap\pi(\boldsymbol{\mu})}\frac{(1+e^{-2it\varepsilon(v)}K^2(v))(1+e^{-2it\varepsilon(v)}/K^2(v))}{(1-e^{-2it\varepsilon(v)})^2}.
\end{aligned}
\tag{182}
$$

*Here the notations are as in Lemma 1 and we have used shorthand notations $K(k) = K_{01;h\gamma}(k)$, $\varepsilon(k) = \varepsilon_{h\gamma}(k)$.*

*Proof.* Inserting two resolutions of the identity in terms of energy eigenstates on either side of $e^{-iH(h,\gamma)t}$ we obtain

$$
{}_{01}^{\mathrm{NS}}\langle\boldsymbol{\lambda}|e^{-itH(h,\gamma)}|\boldsymbol{\mu}\rangle_{01}^{\mathrm{NS}} = \sum_{\boldsymbol{v}}{}_{01}^{\mathrm{NS}}\langle\boldsymbol{\lambda}|\boldsymbol{v}\rangle_{h\gamma}^{\mathrm{NS}}{}_{h\gamma}^{\mathrm{NS}}\langle\boldsymbol{v}|\boldsymbol{\mu}\rangle_{01}^{\mathrm{NS}}e^{-it\mathfrak{E}^{\mathrm{NS}}}\prod_{v\in\boldsymbol{v}}e^{-it\varepsilon(v)}
$$

$$
\begin{aligned}
&= \mathbf{1}_{\sigma(\boldsymbol{\lambda})=\sigma(\boldsymbol{\mu})}e^{-it\mathfrak{E}^{\mathrm{NS}}}\prod_{k\in\sigma(\boldsymbol{\lambda})}e^{-it\varepsilon(k)}(1+K^2(k))\prod_{\mu\in\pi(\boldsymbol{\mu})}(-iK(\mu))\prod_{\lambda\in\pi(\boldsymbol{\lambda})}iK(\lambda)\prod_{p\in\mathrm{NS}_+}\frac{1}{1+K^2(p)} \\
&\times \sum_{\substack{\boldsymbol{v}\subset\mathrm{NS}_+ \\ \cap[\sigma(\boldsymbol{\lambda})\cup(-\sigma(\boldsymbol{\lambda}))]=\emptyset}}\prod_{v\in\boldsymbol{v}}e^{-2it\varepsilon(v)}\begin{cases}\frac{1}{K^2(v)} & \text{if } v\in\pi(\boldsymbol{\lambda})\cap\pi(\boldsymbol{\mu}), \\ -1 & \text{if } v\in\pi(\boldsymbol{\lambda})\perp\pi(\boldsymbol{\mu}), \\ K^2(v) & \text{if } v\notin\pi(\boldsymbol{\lambda})\cup\pi(\boldsymbol{\mu}).\end{cases}
\end{aligned}
\tag{183}
$$

The last line can be rewritten in the form

$$
\prod_{k\in\pi(\boldsymbol{\lambda})\cap\pi(\boldsymbol{\mu})}[1+\frac{e^{-2it\varepsilon(k)}}{K^2(k)}]\prod_{k\in\pi(\boldsymbol{\lambda})\perp\pi(\boldsymbol{\mu})}[1-e^{-2it\varepsilon(k)}]\prod_{\substack{k\notin\pi(\boldsymbol{\lambda}),\pi(\boldsymbol{\mu}) \\ \notin[\sigma(\boldsymbol{\lambda})\cup(-\sigma(\boldsymbol{\lambda}))]}}[1+e^{-2it\varepsilon(k)}K^2(k)].
\tag{184}
$$

Substituting this back in (183) results in the representation given in the Lemma. $\qquad\square$

## B.2 Summation formulas

**Lemma 5** (Andréief identity [128]). *Given two functions $f(\lambda, \mu)$ and $g(\lambda, \mu)$, a set $K$ and two sets of numbers $\{\lambda_i\}_{i=1}^N, \{\mu_j\}_{j=1}^N$ we have the relation*

$$\sum_{k_1 < \cdots < k_N \in K} \det_{i,j}\left[f(\lambda_i, k_j)\right] \det_{i,j}\left[g(k_i, \mu_j)\right] = \det_{i,j}\left[\sum_{k \in K} f(\lambda_i, k) g(k, \mu_j)\right]. \tag{185}$$

*Proof.* See Ref. [1]. □

**Lemma 6** (de Bruijn identity [129]). *Let $f(\lambda, \mu)$ and $g(\lambda, \mu)$ be two functions of two variables, a set $K$ and a set of numbers $\{\lambda_i\}_{i=1}^{2N}$. Define a matrix*

$$\left(A(\boldsymbol{k})\right)_{ij} = \begin{cases} f(k_q, \lambda_j) & \text{if } i = 2q - 1, \\ g(k_q, \lambda_j) & \text{if } i = 2q. \end{cases} \tag{186}$$

*Then the following indentity holds*

$$\sum_{k_1 < \cdots < k_N \in K} \det A(\boldsymbol{k}) = \underset{ij}{\mathrm{pf}}\left[\sum_{k \in K} f(k, \lambda_i) g(k, \lambda_j) - f(k, \lambda_j) g(k, \lambda_i)\right]. \tag{187}$$

*Proof.* Using the definition of the determinant we have

$$\det A(\boldsymbol{k}) = \sum_{\sigma \in \mathfrak{S}_{2N}} (-1)^\sigma f(k_1, \lambda_{\sigma(1)}) g(k_1, \lambda_{\sigma(2)}) \ldots f(k_N, \lambda_{\sigma(2N-1)}) g(k_N, \lambda_{\sigma(2N)}). \tag{188}$$

Then

$$\sum_{k_1 < \cdots < k_N} \det A(\boldsymbol{k}) = \frac{1}{N!} \sum_{k_1, \ldots, k_N} \det A(\boldsymbol{k}) = \frac{1}{N!} \sum_{\sigma \in \mathfrak{S}_{2N}} (-1)^\sigma b_{\sigma(1)\sigma(2)} \ldots b_{\sigma(2N-1)\sigma(2N)}, \tag{189}$$

where

$$b_{ij} = \sum_k f(k, \lambda_i) g(k, \lambda_j). \tag{190}$$

Changing variables to $\sigma = \sigma' \cdot (1, 2)$ we have

$$\sum_{\sigma \in \mathfrak{S}_{2N}} (-1)^\sigma b_{\sigma(1)\sigma(2)} \ldots b_{\sigma(2N-1)\sigma(2N)} = \sum_{\sigma \in \mathfrak{S}_{2N}} \frac{(-1)^\sigma}{2} (b_{\sigma(1)\sigma(2)} - b_{\sigma(2)\sigma(1)}) \ldots b_{\sigma(2N-1)\sigma(2N)}$$

$$= \frac{1}{2^N} \sum_{\sigma \in \mathfrak{S}_{2N}} (-1)^\sigma B_{\sigma(1)\sigma(2)} \ldots B_{\sigma(2N-1)\sigma(2N)}, \tag{191}$$

where $B_{ij}$ is the matrix on the right-hand side in the Lemma. This completes the proof. □

## B.3 Coherent averages

**Lemma 7.** *Let $F[\boldsymbol{q}]$ be a function of $\boldsymbol{q}$, and $f(k)$ a function. We define*

$$\langle F \rangle \equiv \frac{1}{\prod_{k \in \mathrm{NS}_+}[1 + |f(k)|^2]} \sum_{\boldsymbol{q} \subset \mathrm{NS}_+} F[\boldsymbol{q}] \prod_{q \in \boldsymbol{q}} [|f(q)|^2]. \tag{192}$$

*If in the thermodynamic limit $F[\boldsymbol{q}]$ depends on the momenta only through the root density $\rho$, i.e. $\lim_{\mathrm{th}} F[\boldsymbol{q}] = F[\rho]$, then*

$$\langle F \rangle = F[\rho_s] + o(L^0),$$

$$\rho_s(k) = \frac{1}{2\pi} \frac{|f(k)|^2}{1 + |f(k)|^2}. \tag{193}$$

*Proof.* See Ref. [1]. □

**Lemma 8.** *Given two functionals $F[\boldsymbol{q}]$ and $G[\boldsymbol{q}]$, as well as three functions $f(k), g(k), h(k)$, we define*

$$\langle F, G \rangle \equiv \frac{\sum_{\boldsymbol{\lambda}, \boldsymbol{\mu} \subset \mathrm{NS}_+} F[\boldsymbol{\lambda}] G[\boldsymbol{\mu}] \prod_{\lambda \in \boldsymbol{\lambda}} f(\lambda) \prod_{\mu \in \boldsymbol{\mu}} g(\mu) \prod_{\nu \in \boldsymbol{\lambda} \cap \boldsymbol{\mu}} h(\nu)}{\prod_{k \in \mathrm{NS}_+} [1 + f(k) + g(k) + f(k)g(k)h(k)]} . \tag{194}$$

*If $F[\boldsymbol{q}]$ and $G[\boldsymbol{q}]$ depend only on the root density of $\boldsymbol{q}$ in the thermodynamic limit, then*

$$\langle F, G \rangle = F[\rho_1] G[\rho_2] + o(L^{-0}), \tag{195}$$

*with*

$$\rho_1 = \frac{1}{2\pi} \frac{f + fgh}{1 + f + g + fgh}, \qquad \rho_2 = \frac{1}{2\pi} \frac{g + fgh}{1 + f + g + fgh}. \tag{196}$$

*In these equations the root density can be complex.*

*Proof.* It is a generalisation of the proof of Lemma 7 in [1]. Let us first treat the particular case where in the thermodynamic limit $F$ and $G$ depend only on $r$ the number of elements of $\boldsymbol{q}$ divided by $L$. We introduce the generating function

$$\Gamma(\alpha; \beta) = \frac{\sum_{\boldsymbol{\lambda}, \boldsymbol{\mu} \subset \mathrm{NS}_+} \prod_{\lambda \in \boldsymbol{\lambda}} [1 + \frac{\alpha}{L}] f(\lambda) \prod_{\mu \in \boldsymbol{\mu}} [1 + \frac{\beta}{L}] g(\mu) \prod_{\nu \in \boldsymbol{\lambda} \cap \boldsymbol{\mu}} h(\nu)}{\prod_{k \in \mathrm{NS}_+} [1 + f(k) + g(k) + f(k)g(k)h(k)]} . \tag{197}$$

We note that the denominator is such that $\Gamma(0;0) = 1$. By differentiating with respect to $\alpha$ and $\beta$, we see that

$$\langle r^i, r^j \rangle = \partial_\alpha^i \partial_\beta^j \Gamma(0;0) + \mathcal{O}(L^{-1}). \tag{198}$$

Besides, performing the summation on $\boldsymbol{\lambda}, \boldsymbol{\mu}$ we obtain

$$\Gamma(\alpha) = \prod_{k \in \mathrm{NS}_+} \left[ 1 + \frac{\alpha}{L} \frac{f + fgh}{1 + f + g + fgh} + \frac{\beta}{L} \frac{g + fgh}{1 + f + g + fgh} + \frac{\alpha\beta}{L^2} \frac{fgh}{1 + f + g + fgh} \right] (k). \tag{199}$$

From this we find for any $i, j$

$$\langle r^i, r^j \rangle = \left( \int_0^\pi \rho_1(k) \mathrm{d}k \right)^i \left( \int_0^\pi \rho_2(k) \mathrm{d}k \right)^j + \mathcal{O}(L^{-1}), \tag{200}$$

with $\rho_1, \rho_2$ given in the Lemma. As any regular function can be approximated by a polynomial with arbitrary precision provided its degree is high enough, this establishes the result of the Lemma when $F$ and $G$ are functions of $r$ only.

Let us now divide $[0, \pi]$ into $m$ windows $W_k = [\frac{\pi}{m}(k-1), \frac{\pi}{m}k]$ for $k = 1, \ldots, m$, and consider $F[r_1, \ldots, r_m], G[r_1, \ldots, r_m]$ functions of $\boldsymbol{q}$ that in the thermodynamic limit depend only on $r_k$'s, the number of elements of $\boldsymbol{q}$ in $W_k$ divided by $L$. By introducing $\Gamma(\alpha_1, \ldots, \alpha_m; \beta_1, \ldots, \beta_m)$ as in (197) with $\alpha$ replaced by $\alpha_k$ where $k$ is such that $\lambda, \mu \in W_k$, we get similarly

$$\langle r_1^{i_1} \ldots r_m^{i_m}, r_1^{j_1} \ldots r_m^{j_m} \rangle = \prod_{a=1}^m \left( \int_{W_a} \rho_1 \right)^{i_a} \left( \int_{W_a} \rho_2 \right)^{j_a} + \mathcal{O}(L^{-1}). \tag{201}$$

Hence the Lemma holds whenever $F$, $G$ are functions of $r_1, \ldots, r_m$ only. Since any regular functional of $\rho$ can be approximated with arbitrary precision by such a function provided $m$ is large enough, the Lemma holds for general $F[\rho]$ and $G[\rho]$. □

## B.4 Fredholm determinants

**Lemma 9** (Generalized Cramer's rule). *Let $A$ be an $N \times N$ matrix and $x^{i_1}, \ldots, x^{i_k}$ vectors of size $N$. We define $B$ to be the matrix obtained from $A$ by replacing the columns $i_1, \ldots, i_k$ by $x^{i_1}, \ldots, x^{i_k}$. Then*

$$\det B = \det Y \det A, \tag{202}$$

*with $Y$ the $k \times k$ matrix with entries $Y_{i_a, i_b} = y_{i_b}^{i_a}$, where the vector $y^{i_a}$ is a solution to*

$$A y^{i_a} = x^{i_a}. \tag{203}$$

*Proof.* Denoting the columns of $A$ by $C_1, \ldots, C_N$ Eq (203) reads

$$x^{i_a} = \sum_{j=1}^{N} y_j^{i_a} C_j. \tag{204}$$

Using the multilinearity of the determinant, one has

$$\det B = \sum_{j_1, \ldots, j_k = 1}^{N} y_{j_1}^{i_1} \ldots y_{j_k}^{i_k} \det A^{j_1, \ldots, j_k}, \tag{205}$$

where $A^{j_1, \ldots, j_k}$ denotes the matrix obtained from $A$ by replacing the columns $i_1, \ldots, i_k$ by $C_{j_1}, \ldots, C_{j_k}$. Since its determinant is non-zero only if $\{j_1, \ldots, j_k\} = \{i_1, \ldots, i_k\}$, we obtain

$$\begin{aligned}
\det B &= \sum_{\sigma \in \mathfrak{S}_k} y_{i_{\sigma(1)}}^{i_1} \ldots y_{i_{\sigma(k)}}^{i_k} \det A^{i_{\sigma(1)}, \ldots, i_{\sigma(k)}} \\
&= \det Y \det A.
\end{aligned} \tag{206}$$

$\square$

**Lemma 10.** *Let $f(\lambda, \mu)$ be a function of two variables, $J \subset \{1, \ldots, L\}$ a set of $n$ indices, $(a_{ij})_{i,j \in J}$ $n^2$ numbers and $(g_i(\mu))_{i \in J}$, $(h_j(\lambda))_{j \in J}$ $2n$ functions of a single variable. Define an $L \times L$ matrix $A$ by*

$$A_{ij} = \begin{cases} \delta_{ij} + \frac{1}{L} f(\frac{i}{L}, \frac{j}{L}) & \text{if } i, j \notin J, \\ \frac{1}{L} g_j(\frac{i}{L}) & \text{if } j \in J, i \notin J, \\ \frac{1}{L} h_i(\frac{j}{L}) & \text{if } i \in J, j \notin J, \\ \frac{1}{L} a_{ij} & \text{if } i, j \in J. \end{cases} \tag{207}$$

*Then in the limit $L \to \infty$ the following Fredholm determinant representation holds*

$$\det A = \frac{1}{L^n} \det \left[ a_{ij} - \int_0^1 \int_0^1 h_i(\lambda) g_j(\mu) \phi(\lambda, \mu) d\lambda d\mu \right]_{1 \le i, j \le n} \cdot \text{Det}[\text{Id} + f] + o(L^{-n}), \tag{208}$$

*where $\phi$ is the resolvent of the Fredholm equation*

$$\phi(\lambda, \mu) + \int_0^1 f(\lambda, \nu) \phi(\nu, \mu) d\nu = \delta(\lambda - \mu). \tag{209}$$

*Proof.* Using Lemma 9, one has

$$\det A = \det A' \det X. \tag{210}$$

Here $A'$ is the $L \times L$ matrix

$$A'_{ij} = \begin{cases} \delta_{ij} + \frac{1}{L}f(\frac{i}{L}, \frac{j}{L}) & \text{if } i \notin J\,, \\ \frac{1}{L}h_i(\frac{j}{L}) & \text{if } i \in J,\, j \notin J\,, \\ \delta_{ij} + \frac{1}{L}f(\frac{i}{L}, \frac{j}{L}) & \text{if } i,j \in J\,, \end{cases} \tag{211}$$

and $X$ the $n \times n$ matrix whose entry $X_{ij}$ for $i, j \in J$ is the $i$-th element of the solution $x^j$ to the linear system

$$A'x^j = b^j\,, \tag{212}$$

with the vector

$$b_i^j = \begin{cases} \frac{1}{L}g_j(\frac{i}{L}) & \text{if } i \notin J\,, \\ \frac{1}{L}a_{ij} & \text{if } i \in J\,. \end{cases} \tag{213}$$

As $A'$ differs from the matrix with elements $\delta_{ij} + \frac{1}{L}f(\frac{i}{L}, \frac{j}{L})$ by only off-diagonal terms on a finite number of rows in the thermodynamic limit, one has the Fredholm determinant

$$\det A' = \text{Det}[\text{Id} + f] + o(L^0)\,. \tag{214}$$

For $i \notin J$ the system (212) reads

$$x_i^j + \frac{1}{L}\sum_{k=1}^{L} f(\tfrac{i}{L}, \tfrac{k}{L})x_k^j = \frac{1}{L}g_j(\tfrac{i}{L})\,. \tag{215}$$

This equation allows one to describe the entries $x_i^j$ for $i \notin J$ by a function $x^j(\lambda)$ in the thermodynamic limit, since $x_k^j$ for $k \in J$ appears a finite number of times with a factor $\frac{1}{L}$. However, the entry $x_i^j$ for $i \in J$ can be discontinuous in the $i$ direction in the thermodynamic limit. We thus obtain an integral equation for $x^j(\lambda)$

$$x^j(\lambda) + \int_0^1 f(\lambda, \nu)x^j(\nu)\mathrm{d}\nu = \frac{1}{L}g_j(\lambda)\,, \tag{216}$$

whose solution is expressed as

$$x^j(\lambda) = \frac{1}{L}\int_0^1 \phi(\lambda, \mu)g_j(\mu)\mathrm{d}\mu\,, \tag{217}$$

with the resolvent $\phi(\lambda, \mu)$ defined by

$$\phi(\lambda, \mu) + \int_0^1 f(\lambda, \nu)\phi(\nu, \mu)\mathrm{d}\nu = \delta(\lambda - \mu)\,, \tag{218}$$

or equivalently

$$\phi(\lambda, \mu) + \int_0^1 \phi(\lambda, \nu)f(\nu, \mu)\mathrm{d}\nu = \delta(\lambda - \mu)\,. \tag{219}$$

Now, for $i, j \in J$ the system (212) reads

$$x_i^j + \frac{1}{L}\sum_{k=1}^{L} h_i(\tfrac{k}{L})x_k^j = \frac{1}{L}a_{ij}\,. \tag{220}$$

In the thermodynamic limit, this yields for $i, j \in J$

$$x_i^j = \frac{1}{L}\left(a_{ij} - \int_0^1 \int_0^1 h_i(\lambda)g_j(\mu)\phi(\lambda, \mu)\mathrm{d}\lambda\mathrm{d}\mu\right)\,. \tag{221}$$

This concludes the proof of the Lemma.

$$\square$$

### B.5 Miscellaneous

**Lemma 11.** *For $z$ complex and $0 \leq \ell < L$ an integer, we have in finite size $L$*

$$\sum_{v \in \mathrm{R}} \frac{e^{i(\ell+1)v}}{e^{iv} - z} = L \frac{z^\ell}{1 - z^L} \,, \tag{222}$$

*and*

$$\sum_{v \in \mathrm{NS}} \frac{e^{i(\ell+1)v}}{e^{iv} - z} = L \frac{z^\ell}{1 + z^L} \,. \tag{223}$$

*Proof.* Let us start with (222). The left-hand side is a meromorphic function of $z$ with simple poles in $e^{i\mathrm{R}}$, i.e. for $z^L = 1$. The full result can thus be obtained with analytic continuation from $|z| < 1$. In this region, one has

$$\sum_{v \in \mathrm{R}} \frac{e^{i(\ell+1)v}}{e^{iv} - z} = \sum_{m \geq 0} z^m \sum_{v \in \mathrm{R}} e^{i(\ell-m)v}. \tag{224}$$

If $\ell - m$ is a multiple of $L$, then the sum over $v$ is $L$, and otherwise the sum vanishes. Hence

$$\begin{aligned}
\sum_{v \in \mathrm{R}} \frac{e^{i(\ell+1)v}}{e^{iv} - z} &= \sum_{k \geq 0} z^{\ell+kL} L \\
&= L \frac{z^\ell}{1 - z^L} \,.
\end{aligned} \tag{225}$$

Now, to show (223) we write $\mathrm{NS} = \mathrm{R} + \frac{\pi}{L}$ to obtain

$$\sum_{v \in \mathrm{NS}} \frac{e^{i(\ell+1)v}}{e^{iv} - z} = e^{i\ell\pi/L} \sum_{v \in \mathrm{R}} \frac{e^{i(\ell+1)v}}{e^{iv} - z e^{-i\pi/L}} \,. \tag{226}$$

Using (222), we then obtain (223). $\square$

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
