# Peer review of "Out-of-equilibrium dynamics of the XY spin chain from form factor expansion"

_SciPost Physics, doi:SciPost Phys. 12, 019 (2022)_

## Round 1 · Referee Report · Anonymous (Referee 1) · 2021-8-13

Strengths

  1. Important results are obtained.
  2. An interesting method to study the nonequilibrium dynamics is proposed.
  3. The paper is beautifully written.

Weaknesses

No weaknesses.

Report

The article studies the nonequilibrium dynamics of the XY Heisenberg chain. The authors calculate the order parameter expectation value, two-point function (dynamical), and three-point (static) correlation functions. These results certainly deserve attention. The approach based on the use of coherent states is of particular interest. This method strongly simplifies calculations by choosing the most convenient basis. If this approach can be generalized to models with nontrivial interactions, this will be a significant advance in studying the nonequilibrium dynamics of quantum integrable systems.

The paper is beautifully written. All formulas are either deduced in detail or provided with links to original papers. I only noticed two typos (see requested changes).

I highly recommend this article for publication in its present form.

Requested changes

  1. Line above Theorem 1. Double `at different'.
  2. Paragraph between (153) and (154), 2nd line. \sigma^x_\ell should be \sigma^x_j.

---

## Round 1 · Referee Report · Anonymous (Referee 2) · 2021-9-12

Strengths

original new results, opens a new route of thinking about the subject

clear and detailed presentation of results

Weaknesses

no particular weaknesses

Report

The authors consider correlation functions of local operators in a large family of coherent states of the XY chain with time dependent anisotropy $\gamma (t)$ and magnetic field $h(t)$. The coherent states have peculiar properties. They depend on the parameters $\gamma$ and $h$ of the Hamiltonian and on two arbitrary functions $A$ and $f$. Their time evolution can be expressed as an explicit time evolution of these functions. Moreover, Theorem 1 of the manuscript claims that every coherent state with system parameters $\gamma$ and $h$ and functional parameters $A$ and $f$ can be explicitly re-expressed as a coherent states with an arbitrary different set of system parameters $\tilde \gamma$ and $\tilde h$ and new functional parameters $\tilde A$, $\tilde f$ that depend on $\gamma$, $h$, $\tilde \gamma$ and $\tilde h$. This makes it possible to use special values of $\gamma$, $h$, for which the form factors in the Hamiltonian basis are of Cauchy determinant form, in the calculation of matrix elements and overlaps. Building upon these technical achievements the authors are able to express order parameter one-point functions, dynamical two-point functions and equal time three-point function in the thermodynamic limit in terms of Fredholm determinants and Fredholm Pfaffians. The latter objects are numerically efficient and enable to compute the correlation functions at high numerical precision. They will be also amenable to asymptotic analysis in the framework of non-linear steepest descent methods.

For the physical interpretation the authors show that in the long-time limit their coherent state averages relax to a sub-class of so-called generalized Gibbs ensemble averages.

The paper is well-written. The results are presented in sufficient detail such that the readers may verify them on a technical level. As far as I can judge about it all important references are included.

Scope and quality of the paper doubtlessly qualify it for the publication in SciPost Physics.

Here are a few minor remarks on the text. I guess the authors consider even $L$ on page 4. This might be explicitly stated. Below equation (14) on page 6 it should probably read "are given in Lemma 1" instead of "is given in Lemma 1". On the right hand side of equation (60) I would have expected a bold face K under the integral.

Being not so much of an expert on the subject I would have liked the following two point a bit more discussed. 1.) I am familiar with coherent states of Bosons. Are the coherent states of Fermions as used in the paper a standard notion or were they actually introduced in [66,67]? Are they the most general known coherent states in this case, or are they a sub-class? 2.) I wonder in which cases systems can be prepared in a coherent state, in other words, how much physically relevant such coherent state averages are beyond the fact that they can relax to GGE states. I understand that the authors give a nice example in section 4.1, but I wonder, how special this result is and if there are more relevant examples of this kind.

Requested changes

No specific changes required. Please take my comments in the last paragraph of the report as a suggestion of optional amendments.

---

## Round 1 · Referee Report · Anonymous (Referee 3) · 2021-9-15

Strengths

1- well written 2- interesting technique and results 3- correct

Weaknesses

1- results particular to the XY model

Report

In this paper, a technical development is presented for the XY model. Certain states, called Coherent States, are introduced, which are shown to be the correct states to describe quenches in the XY model. In these states, physical quantities of interest are expressed in terms of Fredholm determinants and pfaffians. Such expressions had not been obtained before in the XY model, although other techniques allow one to obtain such types of expressions in the XX model, which possesses U(1) symmetry.

Identifying a family of explicit states that describes the result of quenches, with arbitrary time-dependent model parameters, is something that had been done in the transverse field Ising model before, and this paper adapts the theory to the XY chain.

The expressions obtained for the various physical quantities studied appear to be new, and are certainly very interesting. They open the door for asymptotic analysis.

I think the paper should be published, after the following comments have been briefly addressed.

  1. Page 6: the term “equilibrium physics” can be misleading with respect to previous literature. Certain states of this type have been characterised previously as “non-equilibrium steady states”, for instance those occurring in the “partitioning protocol” or with “domain wall initial state”. For the XY model see [Aschbacher W H and Pillet C-A 2003 Non-equilibrium steady states of the XY chain J. Stat. Phys. 
112 1153] (this is older than GGEs, but the result can be interpreted as a GGE), and more generally [32,33]. In fact the inclusion in (18) of any H_n that is not time-reversal invariant makes the state "non-equilibrium", in a common nomenclature.

  2. There is a well-known technique by which the XY model is “doubled” in such a way that a U(1) symmetry emerges. Normally, by the doubling trick, in the doubled model, order parameter correlation functions may be evaluated using the standard techniques from U(1) invariant models. Can the authors comment on this?

  3. One quantity of interest is the FCS for transport: the exponential of the total transfer of spin, with sum_{j>J} (sigma^z_j(t) - sigma^z_j(0)) instead of the space-sum of the density as in eq 66. Is this accessible by the current techniques? Working this out here would probably be too much, but at least a comment on feasibility would be welcomed.

Requested changes

1- Address the three comments made in the report 2- Typo: “at different at different“ page 5

---

## Round 2 · Referee Report · Anonymous (Referee 3) · 2021-10-31

Report
I thank the authors for briefly commenting. I disagree partly with the understanding in the authors' answers, perhaps I was not clear in my comments.
First, the state obtained in the XY model by Aschbacher and Pillet is a GGE, in the sense the authors discuss. The model, and the resulting state, are homogeneous, and all local observables in the state are independent of time (not: only the initial partioning is inhomogeneous; the homogeneous stationary state obtained is that describing local observables, resulting at long tiems from local relaxation, as considered by the authors). This falls within the set of GGEs the authors discuss. Yet, the state is also out of equilibrium. This is because the GGE is not time-reversal invariant. In the language of Jaynes or Rigol, it contains charges which are not time-reversal invariant. Thus it is out of equilibrium. The more general situation, the nonequilibirum GGEs obtained from the partitioning protocols in integrable models, is discussed in
O.A. Castro-Alvaredo, B.Doyon, and T.Yoshimura, Emergent hydrodynamics in integrable quantum systems out of equilibrium, Phys. Rev. X 6, 041065 (2016)
B. Bertini, M. Collura, J. De Nardis, and M. Fagotti, Transport in
out-of-equilibrium $XXZ$ chains: exact profiles of charges and currents, Phys. Rev. Lett. 117, 207201 (2016)
Of course, Jaynes' or Rigol's description is insufficient, as it does not properly address convergence problems in the infinite sum of charges in integrable models, and this is why one usually resort to other descriptions of GGEs, such as in terms of density of quasiparticles; but in any case, a notion of extensivity, as correctly mentioned by the authors, is retained and is part of the fundamental definition of GGEs.
Second, in my understanding there is no obstruction in using the doubling trick in quantum chains. This is simply about the fact that a U(1) symmetry appears in the doubled model as soon as there is a free fermion description. The free fermion description holds sector by sector, and thus the doubling trick can be used in each sector. It allows one to re-write JW strings of the original model, in terms of U(1) strings of the doubled model. There is no need for bosonisation. But indeed, I do not know how much of this has been done in the literature; I believe this has been at least partly considered in
J. H. H. Perk, Equations of motion for the transverse correlations of the one-dimensional XY -model at finite temperature, Phys. Lett. A 79 (1980) 1–2.
If the authors would like to make adjustments to account for the above this would be good. In any case, these are matters of general understanding, and I believe the paper can be published as it is.
First, the state obtained in the XY model by Aschbacher and Pillet is a GGE, in the sense the authors discuss. The model, and the resulting state, are homogeneous, and all local observables in the state are independent of time (not: only the initial partioning is inhomogeneous; the homogeneous stationary state obtained is that describing local observables, resulting at long tiems from local relaxation, as considered by the authors). This falls within the set of GGEs the authors discuss. Yet, the state is also out of equilibrium. This is because the GGE is not time-reversal invariant. In the language of Jaynes or Rigol, it contains charges which are not time-reversal invariant. Thus it is out of equilibrium. The more general situation, the nonequilibirum GGEs obtained from the partitioning protocols in integrable models, is discussed in
O.A. Castro-Alvaredo, B.Doyon, and T.Yoshimura, Emergent hydrodynamics in integrable quantum systems out of equilibrium, Phys. Rev. X 6, 041065 (2016)
B. Bertini, M. Collura, J. De Nardis, and M. Fagotti, Transport in
out-of-equilibrium $XXZ$ chains: exact profiles of charges and currents, Phys. Rev. Lett. 117, 207201 (2016)
Of course, Jaynes' or Rigol's description is insufficient, as it does not properly address convergence problems in the infinite sum of charges in integrable models, and this is why one usually resort to other descriptions of GGEs, such as in terms of density of quasiparticles; but in any case, a notion of extensivity, as correctly mentioned by the authors, is retained and is part of the fundamental definition of GGEs.
Second, in my understanding there is no obstruction in using the doubling trick in quantum chains. This is simply about the fact that a U(1) symmetry appears in the doubled model as soon as there is a free fermion description. The free fermion description holds sector by sector, and thus the doubling trick can be used in each sector. It allows one to re-write JW strings of the original model, in terms of U(1) strings of the doubled model. There is no need for bosonisation. But indeed, I do not know how much of this has been done in the literature; I believe this has been at least partly considered in
J. H. H. Perk, Equations of motion for the transverse correlations of the one-dimensional XY -model at finite temperature, Phys. Lett. A 79 (1980) 1–2.
If the authors would like to make adjustments to account for the above this would be good. In any case, these are matters of general understanding, and I believe the paper can be published as it is.
Strengths
- Important results are obtained.
- An interesting method to study the nonequilibrium dynamics is proposed.
- The paper is beautifully written.
Weaknesses
No weaknesses.
Report
The authors have corrected typos mentioned in my report. I suggest publishing the paper in its present form.
Requested changes
No changes.

---

## Round 2 · Author Response

Dear Editor and Referees,
We appreciate the very positive reports about our manuscript. Please find below our answers to the referees with the corresponding changes.
Report1: We thank the referee for their very positive report and recommendation to publish our manuscript in its present form.
Report2: We thank the referee for their very positive report, constructive comments, and spotting a number of typos (which we have corrected). We indeed consider even lattice lengths L and now state this explicitly above equation (1). Our replies to the referee’s points (1) and (2) are as follows:
(1) "Coherent state" usually refers to states of the form $e^{b^\dagger}|0\rangle$ with $b^\dagger$ bosonic creation operators. We refer to states (10) as coherent states as they can be viewed as coherent states of bosonic zero-momentum pairs of fermions. We have added a comment below eqn (10) to make this clear. The states defined in this way are indeed a specific class.
(2) As we show in Section 2.4 the state of the system initialised in a ground state of H(h,\gamma) following and then driven out of equilibrium by any variation of the magnetic field and/or the anisotropy can always be written as a coherent state at all times. So their physical relevance is not merely that they relax to GGE: they exactly describe the out-of-equilibrium time evolution of the system at all times in such protocols. Also, the actual experimental realizability of these coherent states is thus equivalent to that of quantum quenches.
Report3: We thank the referee for their positive report and helpful comments. Our replies to the referee’s points (1)-(3) are as follows:
(1) In contrast to the references quoted by the referee we are concerned with a homogeneous system here. Hence the local physics at late times is described by a generalised Gibbs ensemble, which describes an equilibrium state in the sense of Jaynes (under the constraints that the values of the extensive number of conserved quantities are fixed by the choice of initial state). In this view “equilibrium” refers to the fact that local physical observables cease to be time dependent.
In order to avoid any confusion we have added a short comment on p.6 and added a references to the work of Jaynes, and Rigol et al:
E. T. Jaynes, Information Theory and Statistical Mechanics. II, Phys. Rev. 108, 171 (1957)
Relaxation in a Completely Integrable Many-Body Quantum System: An Ab Initio Study of the Dynamics of the Highly Excited States of 1D Lattice Hard-Core Bosons Marcos Rigol,1 Vanja Dunjko,2,3 Vladimir Yurovsky,4 and Maxim Olshanii2
(2) The referee states that "Normally, by the doubling trick, in the doubled model, order parameter correlation functions may be evaluated using the standard techniques from U(1) invariant models.” Our understanding of the situation is at variance with this statement. We are of course well aware that the doubling technique is useful in the continuum limit, but as far as we know the latter is in fact crucial, because Jordan-Wigner string operators reduce to bosonic vertex operators and become simple. On the lattice they are as far as we know difficult to deal with, see e.g. (3.22) in JB Zuber, C Itzykson - Physical Review D15, 2875 (1977). We are not aware of works in the literature that use the doubling trick on the lattice to obtain the kind of results the referee alludes to and we would be grateful for any specific references.
(3) The proposed expectation value would require more work. However we could compute the expectation value of $e^{i\sum_{j>J}\sigma^z_j(t)}e^{-i\sum_{j>J}\sigma^z_j(0)}$ exactly along the same lines as for the dynamical two-point function of \sigma^x. Indeed, the form factor of $e^{i\sum^z_{j>J}\sigma^z_j(0)}$ is a determinant similarly to \sigma^x, so exactly the same calculations would apply.
Best regards,
The authors
We appreciate the very positive reports about our manuscript. Please find below our answers to the referees with the corresponding changes.
Report1: We thank the referee for their very positive report and recommendation to publish our manuscript in its present form.
Report2: We thank the referee for their very positive report, constructive comments, and spotting a number of typos (which we have corrected). We indeed consider even lattice lengths L and now state this explicitly above equation (1). Our replies to the referee’s points (1) and (2) are as follows:
(1) "Coherent state" usually refers to states of the form $e^{b^\dagger}|0\rangle$ with $b^\dagger$ bosonic creation operators. We refer to states (10) as coherent states as they can be viewed as coherent states of bosonic zero-momentum pairs of fermions. We have added a comment below eqn (10) to make this clear. The states defined in this way are indeed a specific class.
(2) As we show in Section 2.4 the state of the system initialised in a ground state of H(h,\gamma) following and then driven out of equilibrium by any variation of the magnetic field and/or the anisotropy can always be written as a coherent state at all times. So their physical relevance is not merely that they relax to GGE: they exactly describe the out-of-equilibrium time evolution of the system at all times in such protocols. Also, the actual experimental realizability of these coherent states is thus equivalent to that of quantum quenches.
Report3: We thank the referee for their positive report and helpful comments. Our replies to the referee’s points (1)-(3) are as follows:
(1) In contrast to the references quoted by the referee we are concerned with a homogeneous system here. Hence the local physics at late times is described by a generalised Gibbs ensemble, which describes an equilibrium state in the sense of Jaynes (under the constraints that the values of the extensive number of conserved quantities are fixed by the choice of initial state). In this view “equilibrium” refers to the fact that local physical observables cease to be time dependent.
In order to avoid any confusion we have added a short comment on p.6 and added a references to the work of Jaynes, and Rigol et al:
E. T. Jaynes, Information Theory and Statistical Mechanics. II, Phys. Rev. 108, 171 (1957)
Relaxation in a Completely Integrable Many-Body Quantum System: An Ab Initio Study of the Dynamics of the Highly Excited States of 1D Lattice Hard-Core Bosons Marcos Rigol,1 Vanja Dunjko,2,3 Vladimir Yurovsky,4 and Maxim Olshanii2
(2) The referee states that "Normally, by the doubling trick, in the doubled model, order parameter correlation functions may be evaluated using the standard techniques from U(1) invariant models.” Our understanding of the situation is at variance with this statement. We are of course well aware that the doubling technique is useful in the continuum limit, but as far as we know the latter is in fact crucial, because Jordan-Wigner string operators reduce to bosonic vertex operators and become simple. On the lattice they are as far as we know difficult to deal with, see e.g. (3.22) in JB Zuber, C Itzykson - Physical Review D15, 2875 (1977). We are not aware of works in the literature that use the doubling trick on the lattice to obtain the kind of results the referee alludes to and we would be grateful for any specific references.
(3) The proposed expectation value would require more work. However we could compute the expectation value of $e^{i\sum_{j>J}\sigma^z_j(t)}e^{-i\sum_{j>J}\sigma^z_j(0)}$ exactly along the same lines as for the dynamical two-point function of \sigma^x. Indeed, the form factor of $e^{i\sum^z_{j>J}\sigma^z_j(0)}$ is a determinant similarly to \sigma^x, so exactly the same calculations would apply.
Best regards,
The authors

---

## Editorial Decision

published